# scRNA-seq reveals persistent aberrant differentiation of nasal epithelium driven by TNFα and TGFβ in post-COVID syndrome

K. D. Reddy[1,2,3,20], Y. Maluje[4,20], F. Ott[4], R. Saurabh[4], A. Schaaf[2], A. Bohnhorst[1,2,3], S. B. Biedermann[1,2,3], J. Pierstorf[1,2], S. Winkelmann[5], B. Voß[5], M. Laudien[5], T. Bahmer[5,6,7], J. Heyckendorf[7,8], F. Brinkmann[1,2], S. Schreiber[8], W. Lieb[9], C. A. Jakwerth[10,11], C. B. Schmidt-Weber[10,11], G. Hansen[12,13,14], E. von Mutius[11,15,16], K. F. Rabe[6,7], A. M. Dittrich[12,13,14], N. Maison[11,15,16], B. Schaub[11,16,17], M. V. Kopp[18], H. Busch[4,21], M. Weckmann[1,2,3,21], A. Fähnrich[4,19,21] ✉ & the ALLIANCE Study Group as part of the German Centre for Lung Research (DZL)

Post-COVID syndrome (PCS) affects approximately 3-17% of individuals following acute respiratory syndrome coronavirus 2 (SARS-CoV-2) infection and poses a potential global health burden. While improved assessment strategies are emerging, mechanistic insights and treatment options remain limited. This study investigates molecular mechanisms underlying PCS using single-cell RNA (scRNA) transcriptomics combined with in vitro validation. scRNA analysis is performed on nasal biopsies from 25 patients with moderate or severe PCS to investigate differential cell types, signalling pathways, and cell-cell communication. Air-liquid interface cultures are used to validate findings, focusing on the TNFα-TGFβ axis. Severe PCS shows reduced numbers of ciliated cells, increased immune cell infiltration, and heightened inflammatory signaling that drives TGFβ and TNFα upregulation, in the absence of a detectable viral load. These changes trigger epithelial-mesenchymal transition, basal cell expansion and a mis-stratified nasal epithelium. In vitro experiments confirm TGFβ and TNFα as causal cytokines promoting ciliated cell loss and increased basal cell abundance. These findings indicate a sustained severe PCS is not driven by ongoing viral load but by immune cell activity and chronic cytokine production. Targeting the TNFα-TGFβ axis may mitigate immune-mediated nasal tissue damage and support epithelium restoration, offering a potential therapeutic strategy for PCS.

Severe acute respiratory syndrome coronavirus 2 (SARS-CoV-2) is the viral agent that causes COVID-19 (coronavirus disease 2019). The pathophysiology and molecular mechanisms of COVID-19 have been studied in detail via epidemiology, multi-omics analyses, and various single-cell and in vitro models[1,2]. The viral tropism of SARS-CoV-2 is determined by the abundance and co-expression of two key cell-surface proteases, such as angiotensin-converting enzyme 2 (ACE2), transmembrane serine protease 2 (TMPRSS2). These proteins are

---

A full list of affiliations appears at the end of the paper. A list of members and their affiliations appears in the Supplementary Information.
✉e-mail: anke.faehnrich@uksh.de

highly expressed in the airway epithelium, forming the first barrier against environmental exposures[3–5]. As such, the multi-ciliated cells of the upper respiratory tract present the first site of SARS-CoV-2 infection.

Although most individuals infected with SARS-CoV-2 typically experience low-grade symptoms and recover within a few weeks, a substantial number continue to cope with long-lasting symptoms[6,7]. This persistent post-infection multisystem condition, often referred to as "Post-COVID Syndrome" (PCS) or "Long COVID," is characterized by symptoms such as fatigue, shortness of breath, and cognitive dysfunction. These symptoms significantly impede an individual's ability to engage in daily activities for extended periods, ranging from months to years[8,9]. Notably, approximately 10–20% of cases across all age groups, including children, are estimated to experience PCS. The WHO classification validates the significance of PCS as a pertinent pathology, as an internationally classified disease (ICD-10 code).

Various hypotheses exist regarding the cause of PCS. Several studies present mechanisms such as immune dysregulation, activation of complement cascades, altered coagulation, tissue injury, neuronal signaling dysfunction, viral reservoirs, systemic inflammation, and T-cell exhaustion[9]. However, gaps exist regarding PCS susceptibility factors, biomarkers, and pathological mechanisms that can provide focal points for treatment options.

The nasal epithelium supports respiratory functions involving air filtration, humidification, and pathogen protection. Basal cells differentiate into various cell types that produce mucus to trap foreign particles and pathogens. SARS-CoV-2 primarily infects ciliated epithelial cells and replicates more efficiently in the nose compared to the lower respiratory tract[10]. The NAPKON-POP study platform has been initiated in Germany to facilitate population-based PCS studies within the general population and hosts the COVIDOM study[11]. The platform classifies PCS severity based on enduring symptoms that impact the quality of life[8]. While this approach, based on patient-reported symptoms, is beneficial for clinical categorization, associations with specific molecular mechanisms have yet to be identified.

As such, to better understand the pathological mechanisms of PCS, we hypothesize a residual disease state in the nasal epithelia in post-COVID patients. For this, we extended the NAPKON examination protocol and obtained nasal biopsies from study participants with moderate and severe PCS. We aimed to identify changes in cellular abundance and interactions that occur post-acute SARS-CoV-2 infection that differ between distinct clinical presentations of PCS.

Here, we show that a severe post-COVID syndrome (PCS) phenotype is characterized by immune-mediated damage to the nasal epithelium, occurring independently of viral persistence. Through single-cell RNA transcriptomics on nasal biopsies from patients with moderate to severe PCS, we observed significant changes in cell-type composition, signaling pathways, and communication between cells. Severe PCS is associated with a decrease in proximal ciliated cells, an increase in immune cell infiltration, and heightened inflammatory signaling, particularly involving TNFα and TGFβ. This inflammatory response drives epithelial-mesenchymal transition and the expansion of basal cells. In vitro validation using air-liquid interface cultures shows that TGFβ and TNFα are critical factors contributing to improper basal cell differentiation and the malformation of the airway mucosa. These findings suggest that targeting the TNFα-TGFβ pathway may provide a therapeutic strategy to restore epithelial integrity in patients with PCS.

## Results

### Post-COVID symptom severity correlates with multi-organ issues and increased risk of respiratory comorbidities

We selected 33 patients (whose entire metadata was available) recruited from the NAPKON study cohort ($n = 1270$). Using the PCS questionnaire developed previously[8], these 33 participants were classified into mild ($n = 4$), moderate ($n = 11$), and severe ($n = 18$) PCS based on long-term symptom complexes. Supplementary Fig. S1 contains a breakdown of the number of samples selected from the NAPKON cohort. The threshold for each PCS group classification was determined at mild PCS < 10.75, moderate 10.75–25.25, and severe PCS > 25.25. Using a curette, the nasal epithelium was collected from the anterior and medial heads of the middle turbinate (Fig. 1). All patient groups had a comparable age, with no sex imbalance between the moderate and severe PCS groups. Severe PCS patients reported more joint and muscle pain, and skin complaints compared to moderate PCS patients. Notably, severe PCS patients also present with higher rates of breathing problems and symptoms of infections (Supplementary Table S1). This indicates an apparent increased disruption of respiratory function in severe PCS.

### scRNA-seq analysis identifies differently stratified nasal epithelia with PCS severity

To examine differences in cellular composition, pathway activities, and cell-cell communication in the nasal epithelia of patients with PCS, we conducted single-cell RNA sequencing (scRNA-seq) on 33 patient samples. Four samples did not pass initial quality control. The remaining 29 samples yielded 56,624 cells, each with less than 25% mitochondrial gene content and more than 200 genes detected. The group of patients with mild PCS ($n = 4$), consisting solely of males, was statistically smaller and significantly imbalanced. Consequently, these

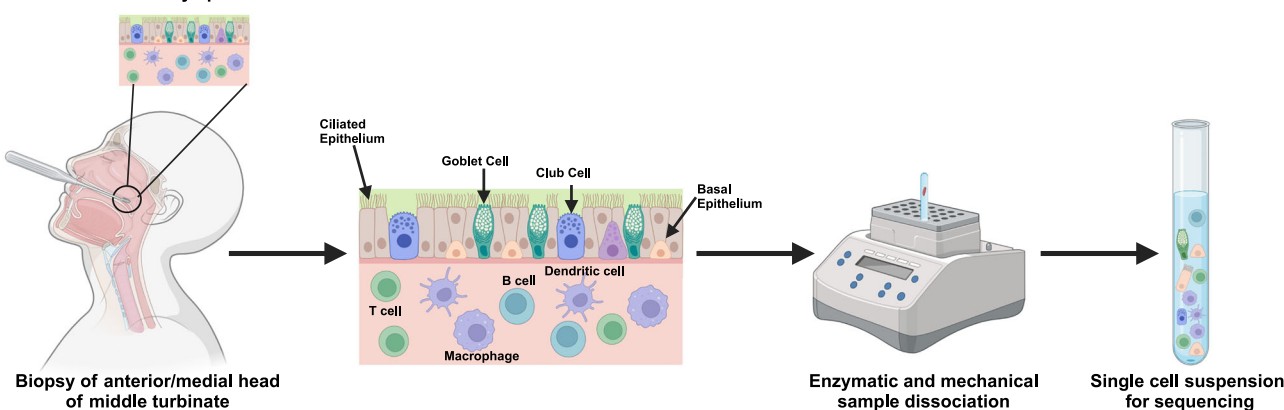

**Fig. 1 | Schematic representation of sample collection and processing.** Nasal samples were collected using a currette, capturing the epithelium and local immune population. Samples were processed and dissociated to create a single-cell suspension for downstream single-cell RNA sequencing analysis. Created in BioRender https://BioRender.com/tvlyurc.

samples were excluded to ensure statistically robust and meaningful comparisons.

Eventually, our analysis yielded 15,014 cells from moderate PCS ($n = 11$) and 29,680 cells from severe PCS ($n = 14$) patients. Final quality checking was completed using the *Seurat* pipeline[12] to account for inherent variation between the patient samples (Supplementary Figs. S2 and S3). Samples were integrated using the *FindIntegrationAnchors* function to enable cross-patient cluster comparison. Subsequently, using the *Seurat* pipeline[12], we identified 17 unique cell clusters, based on known gene expression markers for epithelial cells of the conducting airways. These include keratin 5/7 (*KRT5/7*) for proliferating, differentiating basal and mucosal cells, tubulin beta 4B class IVb (*TUBB4B*) and Coiled-coil domain containing 153 (*CCDC153*) for ciliated cells, and cystic fibrosis transmembrane conductance regulator (*CFTR*) for ionocytes[13]. The annotation confirmed the presence of nasal epithelial and immune cells (Fig. 2A, B), such as basal cells, myeloid-dendritic cells (*CD1C* and *FCER1G*), and T cells (*TRBC2*). The marker expression profile for the myeloid-dendritic cells indicates this cluster is likely a mixture of dendritic cells, monocytes, and macrophages. In addition to the previously mentioned markers, we did not detect any significant expression of *CD20* or *IGHD*, which are indicative of B-cells. Supplementary Table S2 presents a list of differentially expressed genes and canonical markers for cluster annotation. For subsequent analyses, we collapsed the subclusters of differentiating basal, goblet, ciliated, and proximal ciliated cells to create a total of 12 clusters based solely on cell type (Fig. 2A, B).

Previous studies identified the nasal epithelium as the acute SARS-CoV-2 infection site, causing altered nasal epithelial composition[14]. Analysis of SARS-CoV-2 genetic sequences revealed a lack of viral presence in these patients. Further, analysis of *IL6* and *CXCL8* expression corroborated a lack of active viral infection (Supplementary Fig. S4), as *IL6* and *CXCL8* are upregulated during acute infection[15]. This indicates viral clearance and the absence of an active inflammatory response to infection. Nonetheless, a significant change in the relative airway cell composition occurs with PCS severity (Fig. 2C). Proximal ciliated cells, their precursors, and deuterosomal epithelial cells decreased in relative abundance with increasing PCS scores. Both DA-seq and scProportion methods (using a calculated threshold; Supplementary Fig. S5) indicated a $-0.5$ log$_2$-fold change (FC) of proximal ciliated and mucous cells (FDR < 0.05), and a significant reduction of deuterosomal epithelial cells ($-1.2$ log$_2$FC, FDR < 0.05) in severe compared to moderate PCS (Fig. 2D). Conversely, T-cell (1.5 log$_2$FC), basal cell (1.3 log$_2$FC), and myeloid-dendritic cell (0.6 log$_2$FC) abundance increased in severe PCS (Fig. 2D). Hereby, we report that proximal ciliated cells are depleted, whilst basal epithelium, T cells, and myeloid-dendritic cells are increased in severe PCS compared to moderate PCS (Supplementary Fig. S6 and Table S3). Cellular deconvolution of bulk-transcriptome sequencing of healthy control and asthmatic nasal epithelium (ALLIANCE cohort) demonstrates that severe PCS patients report fewer ciliated cells (33% vs. 25%; Supplementary Fig. S7).

scRNA-seq analysis indicates a persistently altered differentiation and incomplete formation of the nasal mucosa, contributing to increased susceptibility to future pathology and the promotion of a PCS phenotype. Due to this apparent divergence from a regular nasal epithelium, we hypothesized that the protective function of the nasal epithelium against pathogens and irritants is subsequently diminished. As such, we extracted information from the TriNetX network of healthcare organizations[16], to explore whether the presence of PCS is linked to an increased risk of infection throughout the respiratory tract. Patients with PCS, according to ICD-10 codes, returned 52,833 cases and 51,310 propensity-matched controls. We identified 22 associated comorbidities with increased odds and hazard ratios (OR and HR > 1; Fig. 2E, Supplementary Table S4). Of those comorbidities, chronic respiratory failure, interstitial pulmonary disease, asthma, and pulmonary fibrosis are associated with dysmorphic mucosa. Viral

pneumonia (OR and HR > 4), acute bronchiolitis (Fig. 2E), may originate from the lack of protective function of the nasal epithelium.

## Aberrant nasal epithelial composition is driven by TNFα and TGFβ signaling in the immune cell compartment, affecting basal epithelial cell differentiation

To investigate the signaling pathways underlying differences in cell composition between moderate and severe PCS, we examined the pathways enriched upstream of the differentially regulated genes[17]. Severe PCS reported inflammation-related enrichment (TNFα, TGFβ, NF-κB; Fig. 3A). This, combined with increased myeloid and T-cell abundance (Fig. 2D), indicated an activated immune response in severe PCS. The PI3K pathway, enriched in moderate PCS, implicates reduced cell proliferation. Conversely, in severe PCS, TGFβ signaling is enriched, whilst cellular migration pathways are enriched in moderate PCS, as indicated by PI3K enrichment signaling (Fig. 3A).

Using *CellChat*[18], we deduced paracrine cell-cell communication strength and direction via differential expression of known receptor-ligand gene pairs in severe vs. moderate PCS. We observe that basal proliferating cells and myeloid cells form a communication nexus in severe PCS (Fig. 3B, top left). Comparatively, in moderate PCS, mucous and basal cells send signals, whilst T cells receive cellular signals in moderate PCS (Fig. 3B, bottom right). In severe PCS, basal proliferating cells were predicted as strong senders of cell-cell signals (Fig. 3D), whilst myeloid-dendritic cells are primarily receivers of cell-cell signals in moderate and severe PCS (Fig. 3C, D). Myeloid-dendritic cells receive input from all cell types irrespective of PCS severity yet specifically send signals to basal proliferating and T cells in severe PCS (Fig. 3B–D).

Next, we quantified changes in signaling pathways between moderate and severe PCS. Extracellular matrix (ECM) and cell receptor pathways remained unchanged (Fig. 3E, F). Interestingly, the IGF pathway (indicating cell differentiation[19]) was absent in severe PCS, corroborating the indicated aberrant epithelial cell differentiation (Fig. 3E). Instead, cell-cell communication strength in severe PCS was dominated by cell-cell adhesion (CD46, CD99, JAM), immune signaling (GALECTIN), and cell growth and survival (MK, PROS1; Fig. 3F). The macrophage migration inhibitory factor (MIF) was enriched in severe PCS ligand-receptor interactions (Fig. 3G). MIF is an inflammatory cytokine associated with TNFα and TGFβ production[20,21]. MIF signaling via CD74, CXCR4, and CD44 receptors activates downstream NF-κB, MAPK, and AKT pathways, regulating immune responses, cell proliferation, and cell survival[22,23]. *MIF* and *CD74* were highly expressed across all cell types irrespective of PCS severity (Supplementary Fig. S8A, B). We observed increasing *CXCR4* (myeloid) and *CD44* (myeloid and basal cells) expression with PCS severity (Supplementary Figs. S8C, D). Accordingly, *CellChat* revealed an enrichment of the MIF-(CD74-CXCR4) axis in severe PCS myeloid cells (Fig. 3G), while basal proliferating cells indicate enrichment of signals via CD44 in severe PCS patients (Supplementary Fig. S9).

## Severe PCS indicates enrichment for TNFα and TGFβ pathways with divergent basal epithelial cell differentiation trajectories compared to moderate PCS

Analysis of PROGENy pathway enrichment per cell type revealed an enrichment of EGFR and NF-κB in both myeloid-dendritic cells and T cells (Fig. 4A), confirming the downstream effects of CD74 activation[22,23]. Additionally, PROGENy predicted the upstream activity of TNFα (Supplementary Fig. S10). We also found increased expression of *TNF* and *TGFβ1* in myeloid-dendritic and T cells with PCS severity (Fig. 4B). *TNF* expression was observed primarily in the immune cell compartments, likely driven by the NF-κB pathway. *TGFβ1* expression was increased in immune and basal epithelial cells (Fig. 4B). In the former, *TGFβ1* expression is putatively caused by upstream TGFβ signaling, whilst the latter is driven by WNT or MAPK pathway activation

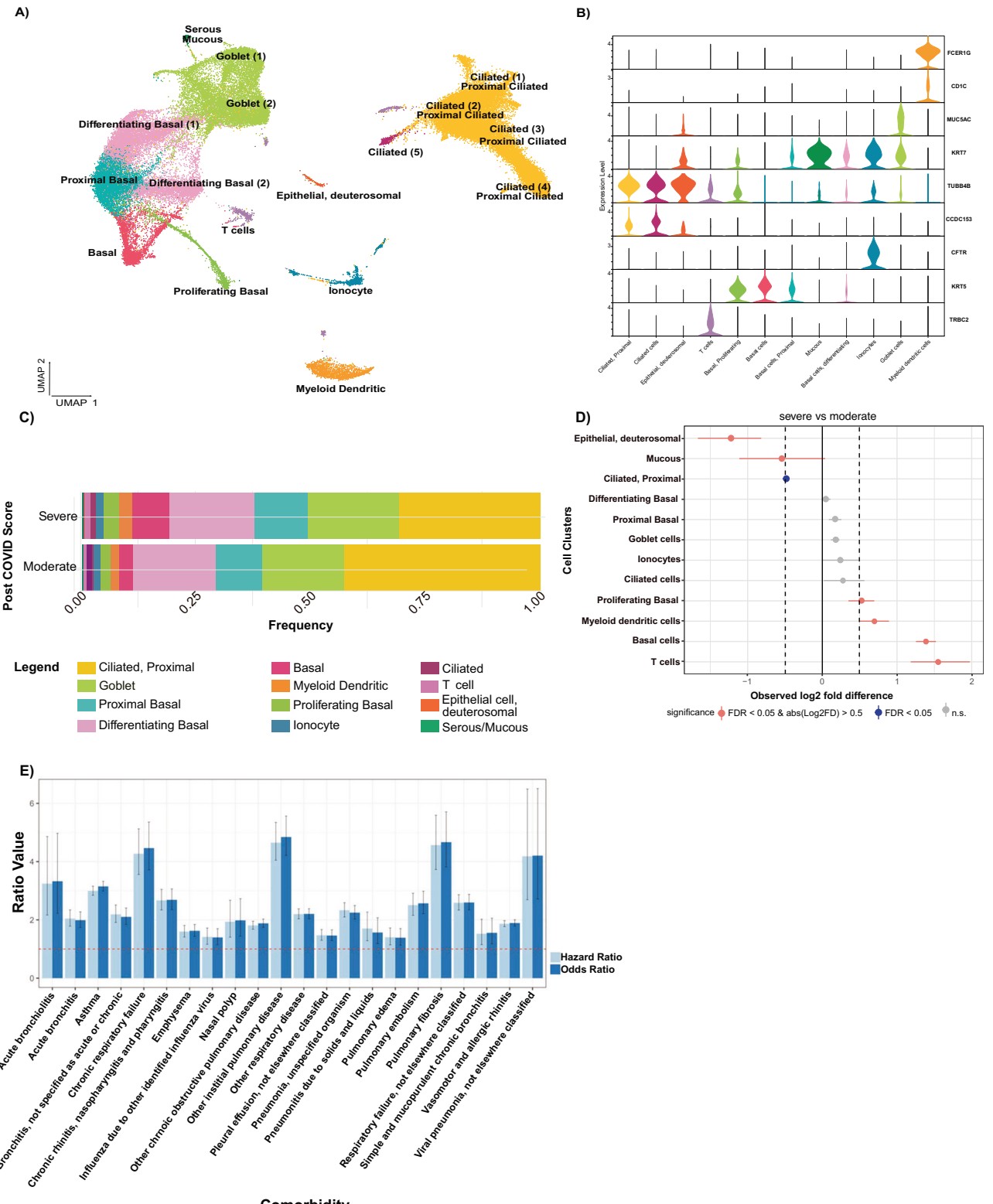

(Fig. 4A, B). All cell types demonstrated increased TNFα and TGFβ receptor expression, which may be caused by increased production of the relevant cytokines (Supplementary Fig. S11).

To check whether TGFβ signaling leads to EMT (important for the nasal epithelium remodeling during inflammation[24]), we performed a Gene Set Enrichment Analysis (GSEA)[25]. Using the human 'Hallmark Gene Set'[25], GSEA confirmed TNFα, NF-κB, and EMT as the most upregulated pathways in the nasal mucosa of severe PCS patients

(Supplementary Fig. S12). Pseudo-time trajectory analysis further investigated our hypothesis of EMT and altered patterns of epithelial cell fate. Figure 4C–E depicts the trajectories of nasal epithelial cell differentiation within moderate and severe PCS patients. The number of edges in moderate (247) and severe PCS (327) trajectories indicates a more branched pseudo-time trajectory with multiple endpoints for severe PCS. This is reflected by a lower centrality in moderate ($8.1 \times 10^{-3}$) vs severe PCS ($3.1 \times 10^{-3}$). Therefore, a more divergent

**Fig. 2 | scRNA-seq analysis of the cellular composition of nasal samples from moderate and severe PCS patients. A** Cell-type annotated UMAP plot of all integrated samples. 17 distinct clusters were detected by cluster gene signatures. **B** Violin plots of marker gene expression (*ln*-transformed counts per million) of the 17 distinct clusters collapsed to 12 common cell types found in the conducting airways. **C** Stacked bar plots of cell cluster frequency in moderate and severe PCS patients. **D** Relative differences in cell cluster proportions between moderate and severe PCS groups. Each point represents the observed $\log_2$ fold-difference in cell-type proportion (severe vs. moderate). Statistical significance was assessed using permutation testing with FDR correction. Error bars denote the 95% confidence intervals (2.5th and 97.5th percentiles), calculated from 1000 permutations.

Significance is color-coded: red (FDR < 0.05 & $|\log_2FD| > 0.5$), blue (FDR < 0.05), and gray (not significant). Vertical black dashed lines indicate the $\pm 0.5 \log_2FD$ thresholds. Group sizes were moderate (11 samples, 15,014 cells) and severe (14 samples, 29,680 cells). **E** Hazard and odds ratios of nasal and respiratory diseases significantly increased in post-COVID patients. Each ratio is plotted in pairs, where light blue and dark blue indicate the hazard and odds ratios. Error bars represent the 95% confidence intervals. The dotted horizontal line at 1.0 indicates no effect; values above this line signify an increased risk or association with the outcome. Specifically, an odds ratio >1 indicates greater odds of the outcome, while a hazard ratio >1 indicates a higher rate of the outcome over time. A total of 52,833 post-COVID cases and 51,310 controls were included in the analysis.

differentiation route from basal epithelial to ciliated cells exists in severe PCS.

Taken together, our analysis shows that basal and dendritic myeloid cells are integral communicators across worsening PCS. However, in severe PCS, basal proliferating cells feature a TGFβ-driven response, with myeloid cells maintaining an inflammatory profile. This inflammation is triggered through MIF and propagated by TNFα signaling, leading to the activation of TGFβ and EMT in severe PCS nasal epithelium.

### TGFβ and TNFα exposure during nasal air-liquid interface causes aberrant differentiation of basal epithelial cells

The PCS scRNA-seq revealed increased TGFβ and TNFα signaling, which we hypothesize drives the reported aberrant nasal epithelium stratification in severe PCS. In our data, differentiating basal and goblet cells were the primary recipient cells for TGFβ and TNFα in severe PCS (Fig. 4A). To investigate these prominent cytokines as causal factors in the reduction of ciliated cells, we exposed basal nasal epithelial cells (NECs) to an air-liquid interface (ALI) model with or without TNFα and TGFβ stimulation either alone or in combination (see workflow in Supplementary Fig. S13). NECs underwent scRNA-seq, and after quality-checking by filtering cells with less than 25% mitochondrial gene content and more than 200 genes, a total of 22,855 cells remained across all conditions. Six distinct clusters were identified (Fig. 5A, Supplementary Fig. S14), with some clusters identified as stimulation-specific (Fig. 5B). TGFβ exposure promoted increased abundance of differentiating basal cells, with reduced ciliated cell abundance compared to control (PBS) conditions (Fig. 5C; Supplementary Fig. S15; Supplementary Table S5). Comparatively, TNFα exposure resulted in greater basal cell differentiation towards ciliated cells and the secretory type cells (Fig. 5C). When combined, TGFβ and TNFα caused an amelioration of multiple cell fates, with basal cells apparently starting to differentiate but never committing to a terminal cell state. Although some epithelial-mesenchymal transition (EMT) towards fibroblast/myofibroblast phenotype was reported (Fig. 5C).

Upstream pathway enrichment analysis across treatment conditions highlights activation of VEGF, hypoxia, p53, JAK-STAT, and EGFR pathways, and most prominently combined TGFβ + TNFα stimulation (Supplementary Fig. S16). These findings closely mirrored the distinct pathway activation profiles observed between the PCS moderate vs PCS severe samples in Fig. 3A. Similarly, GSEA analysis indicates strong EMT pathway enrichment in TGFβ-stimulated cells, whilst oxidative phosphorylation was reduced in both TGFβ- and TNFα-alone (Fig. 5D). In addition, G2M and E2F factors are significantly enriched in PBS-treated samples only, highlighting specific dysregulation of normal cell cycle progression induced by TNFα and TGFβ. This reflects the in vivo results, where we observed altered differentiation of basal epithelial cells, with reduced proliferation signals. Differences between the pathway enrichment profiles are likely due to the complexity of the cellular population analyzed from the patient samples. Stimulation with only TGFβ during differentiation partially replicated the epithelial phenotype seen in severe PCS patients, with this effect amplified by combined TGFβ/TNFα stimulation.

In conclusion, the air-liquid interface in vitro models of TGFβ and TNFα exposure mimicked the reduced cellular differentiation, which was indicated in the severe PCS patient's nasal mucosa. Our model demonstrates that TGFβ causes a loss of basal epithelial differentiation towards ciliated cells, which is exaggerated by co-exposure with TNFα, as illustrated in Fig. 6. This corroborates what we observe on a transcriptional level in our PCS cohort.

## Discussion

In this study, we observed an apparent malformation of pseudo-stratified nasal mucosa in severe PCS patients compared to moderate PCS patients. This was characterized by reduced ciliated epithelium and increased immune cell presence in severe PCS. The prolonged presence of myeloid and T cells in severe PCS was associated with a persistent inflammatory response via MIF-CD74 signaling and increased expression of TNFα and TGFβ. The proposed effects of this perpetuating inflammatory response and EMT of basal epithelial cells were reinforced in vitro by TGFβ combined with TNFα exposure, impairing basal cell differentiation into ciliated cells using an air-liquid interface model. This proposed mechanism for PCS may partly explain the multitude of respiratory comorbidities experienced by PCS patients (TriNetX), such as interstitial pulmonary disease, pulmonary fibrosis, viral pneumonia, bronchiolitis, and asthma. Thus, the persistence of infection-like symptoms combined with a poorly reconstituted respiratory epithelium (identified in the nose) potentially prolongs poor health outcomes for severe PCS patients.

The incidence of PCS varies from 10–30% among non-hospitalized cases to 50–70% among hospitalized cases. However, clear diagnostic criteria are still lacking, making precise estimates difficult[26,27]. PCS affects individuals of all ages, with the highest frequency in non-hospitalized patients between 36 and 50 years[9,27]. Thus, strong health concerns persist despite the COVID-19 pandemic diminishing[27]. Here, we have performed a single-cell transcriptome analysis of nasal epithelial biopsies from a cohort of 25 patients with moderate or severe PCS. We hypothesized that long-lasting structural changes in the nasal mucosa distinguish severe PCS patients, causing the perceived worsened symptom load[14].

scRNA-seq revealed a persistence of myeloid and T cells and latent activation of the MIF-CD74 axis in severe PCS. These data align with previous findings of circulating dendritic cells (DCs) and monocytes increased for six months after severe COVID-19 infections[28]. Roukens et al. detected CD8+ T cells persisting in the nasal mucosa for at least two months after SARS-CoV-2 viral clearance[28]. A recent study has linked post-COVID exercise intolerance to resident immune cell-induced artery remodeling in skeletal muscle despite viral clearance, suggesting long-lasting tissue remodeling is a PCS feature[29]. Immune cell clearance is impeded in PCS and potentially prolonged inflammation-promoting paracrine cell-cell communication via MIF-CD74 and subsequent expression of TNFα and TGFβ. Inhibition of MIF and CD74 resulted in the loss of oxidative stress protection, leading to increased inflammatory cytokine production, apoptosis, and higher mortality rates in animal models[30]. CD74 signals through the ERK and Akt pathway activation[31]. The critical function of CD74 is to stabilize

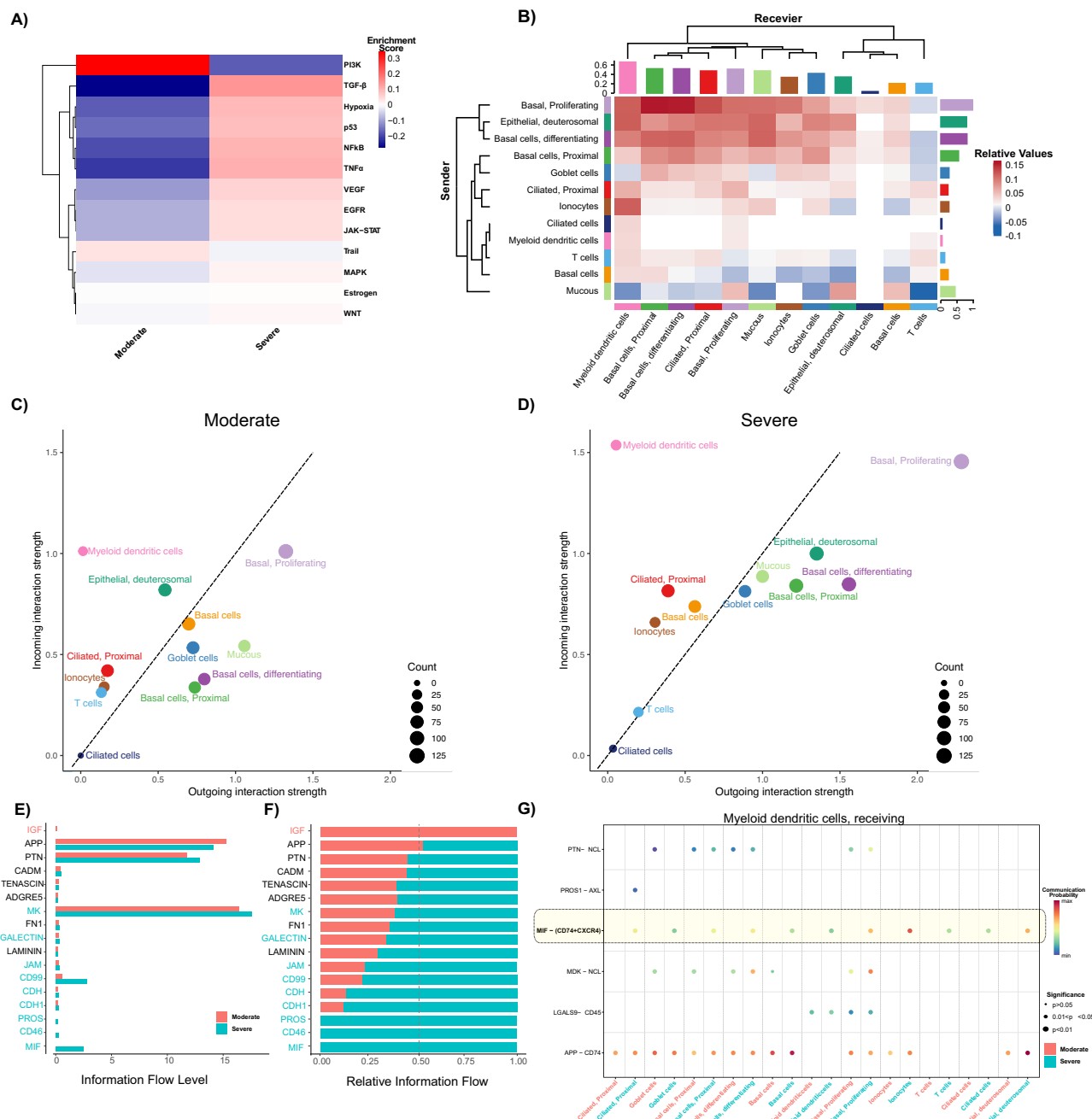

**Fig. 3 | Enriched pathological pathways and cell-cell interactions with PCS severity. A** Relative PROGENy pathway enrichment in moderate and severe PCS. Red indicates positive, whilst blue indicates negative enrichment for severe PCS. **B** Heatmap of the signaling pathway enrichment contributing to outgoing or incoming communication. The color bar indicates the relative signaling strength in PCS severity; red = increased in severe PCS, and blue = decreased in severe PCS. The solid-colored bars on the x- and y-axes indicate the sum of the incoming (x-axis) or outgoing (y-axis) signaling strength for each cell type. Comparison of total incoming signaling strength vs. total outgoing signaling strength across cell populations in moderate (**C**) and severe (**D**) PCS. Dot size is proportional to the number of outgoing and incoming inferred links associated with each cell population group. Dot colors indicate different cell population groups. All significant pathways (accumulated *p*-value < 0.05) are presented as absolute (**E**) or relative (**F**) information flow, ranked based on differences in the overall information flow between moderate (red) and severe (cyan) patients. The overall information flow of

a signaling network was calculated by summarizing all communication probabilities in that network. Pathways colored red are enriched in moderate, those colored cyan are enriched in severe, and black are equally enriched in both conditions. **G** Significant ligand-receptor pair interactions are represented from all other cell groups to myeloid-dendritic cells in moderate (red) and severe (cyan) PCS. The dot color and size represent the calculated communication probability and p-values, respectively; *n* = 25. Empty space means the communication probability is zero. *p*-values are computed from a one-sided random permutation test (100 permutations). GF insulin-like growth factor, APP amyloid beta precursor protein, PTN pleiotrophin, CADM cell adhesion molecule, ADGRE5 adhesion G protein-coupled receptor E5, MK midkine, FN1 fibronectin 1, JAM junction adhesion molecule 2, CDH cadherin-1, PROS protein S, MIF macrophage migration inhibitory factor, NCL nucleolin, AXL receptor tyrosine kinase, CXCR4 CXC motif chemokine receptor 4, LGALS9 galectin 9.

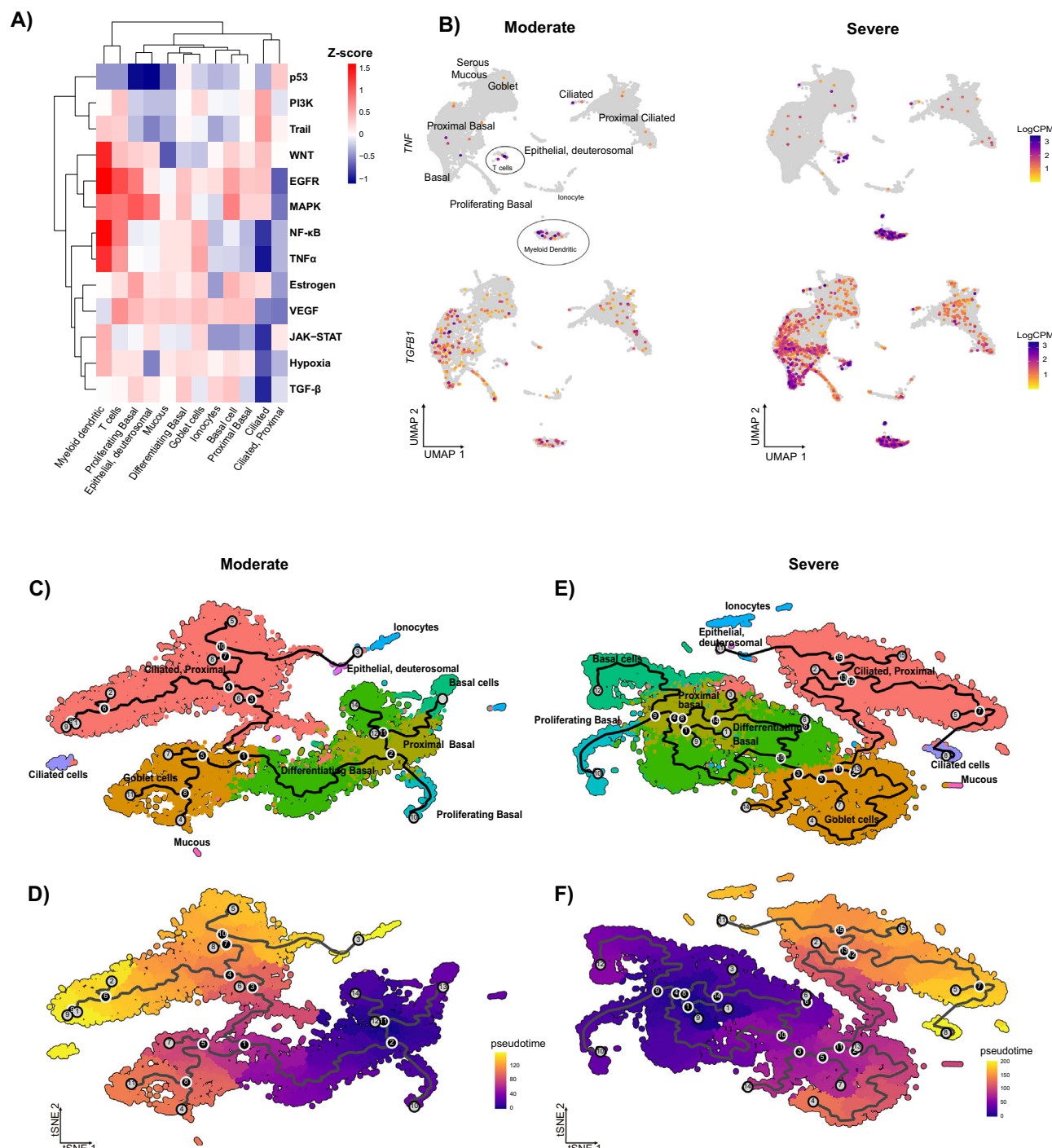

**Fig. 4 | Presentation of cellular expression of *TNFα* and *TGFβ* and altered basal cell differentiation trajectory. A** Heatmap of PROGENy pathway enrichment, stratified by cell type; red indicates positive, whilst blue indicates negative enrichment in severe PCS. **B** UMAP of *TGFβ1* and *TNF* expression; purple = increased expression, and gray = no expression. Gene expression is quantified as log₂Counts per million (CPM). **C–F** Pseudo-time analysis of nasal epithelial cell differentiation in moderate (**C** and **D**) and severe (**E** and **F**) PCS patients. **C** and **E** t-SNE of cell populations. **D** and **F** t-SNE plot with overlay of pseudo-time progression of cell development; purple = earlier in trajectory, and yellow = later in trajectory. Black lines represent the most likely path of cell maturation over the pseudo-time trajectory; *n* = 25.

the presentation of antigens for T cells[20]. When combined with MIF, it may drive the persistent activation of pro-inflammatory pathways in severe PCS[32]. In addition, latent production of TGFβ promotes continued activation of the MIF-CD74 axis via the ERK pathway, generating a self-perpetuating cycle in severe PCS patients[21]. Severity might be aggravated by the appearance of a new immune cell subset linked to two MIF co-receptors, CXCR4 and CD44. Both receptors mediate

differential MIF responses related to immune cell infiltration[31,33] and have been implicated in COVID-19 and acute lung injury[33,34].

Our data also indicates fibronectin induced by TGFβ stimulation is enriched in severe PCS. Fibronectin is well described to induce epithelial proliferation and differentiation[35]. Conversely, laminin B1 is produced more in moderate PCS, has been implicated in modulating correct lung development by promoting epithelial cell adhesion,

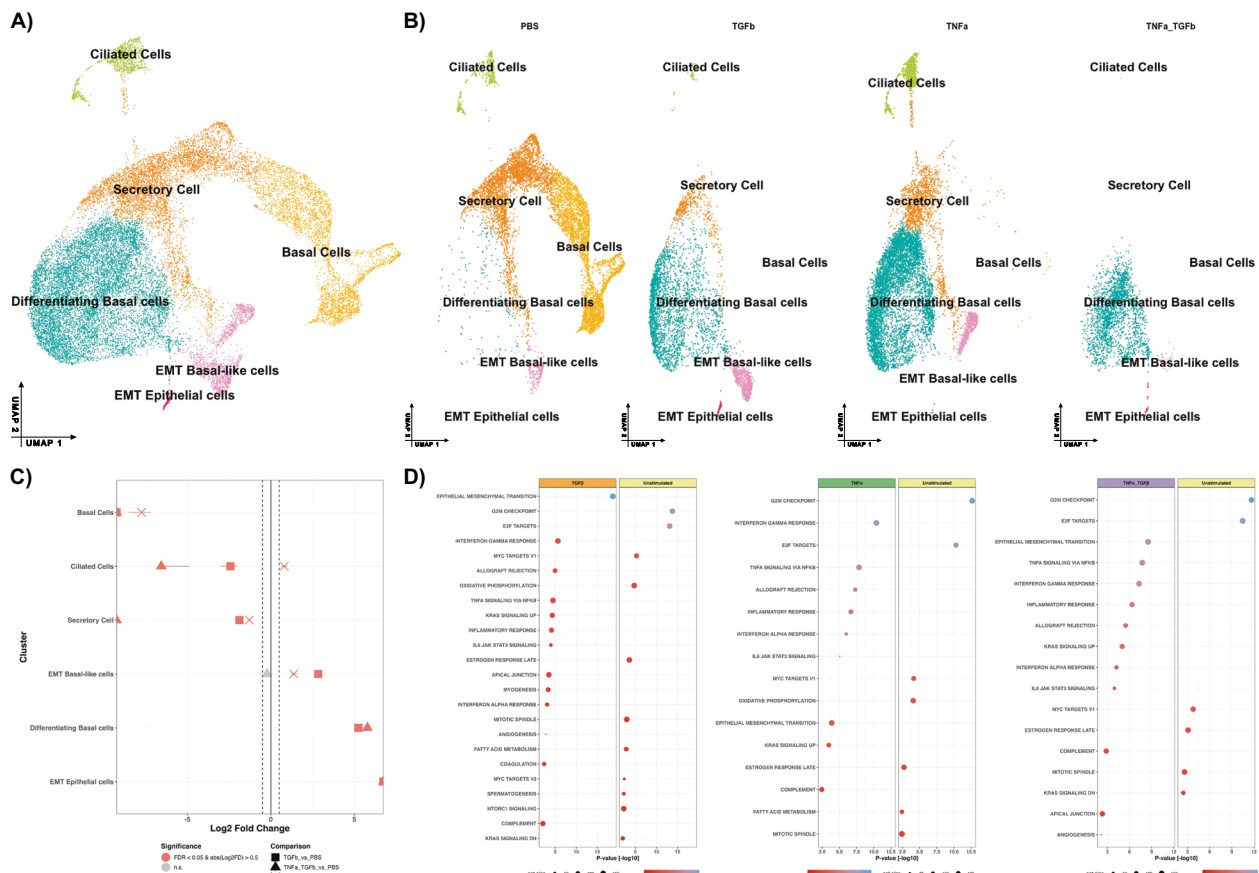

**Fig. 5 | scRNA-seq analysis of ALI cultures stimulated with TGFβ or TNFα alone or in combination. A** UMAP of integrated scRNA-seq of primary nasal epithelial cells differentiated over 28 days across all stimulation conditions. **B** UMAP of individual stimulation conditions PBS, TNFα (10 ng/ml), TGFβ (1 ng/ml), and TNFα with TGFβ combined. Individual cell populations are identified for each condition based on canonical expression markers (Fig. S17). **C** Relative differences in cell proportions for each cluster across all conditions. Red colored shapes indicate a false discovery rate (FDR) < 0.05 and log2 fold-difference >0.5 compared to PBS control conditions; Squares (■) = TGFβ vs PBS, triangle (▲) = TNFα with TGFβ vs PBS, and a cross (✕) = TNFα vs PBS. **D** GSEA pathway enrichment analysis stratified by treatment conditions compared to PBS control. Color of the dots indicates significance, with red representing a smaller log-$p$-value and blue indicating higher. Dot size is proportional to the number of genes identified within each pathway; larger dots indicate gene sets with more associated genes for that pathway; $n = 2$ donors for each condition.

migration, and proliferation[36–38]. These ECM changes, combined with a hypothesized increased deposition of ECM-bound TGFβ[39], could cause 'crowding' of basal epithelial cells followed by anoikis (luminal cell extrusion) as evident by the reduced activation of the PI3K pathway in severe PCS[40]. This, combined with potentially localized TNFα exposure of differentiating basal cells from myeloid and T cells, reducing differentiation into ciliated cells, may cause the aberrant restitution airway epithelium. This may indicate systemic restorative problems in the recovery from acute SARS-CoV-2 infection, contributing to PCS severity.

Considering this within the context of our ALI model, it is possible that the blocking of the TGFβ-receptor could enable the resolution and reparation of the airway epithelium during PCS. Figure 6 indicates that TGFβ may be bound to the ECM, primarily impacting the initial differentiation of basal epithelial cells. Therefore, blocking the initiation of this signaling cascade may enable correct differentiation and formation of the airway epithelium. This is evidenced by previous findings showing that blocking TNFα or TGFβ can improve patient outcomes. Interestingly, chronic inflammation in nasal tissue has been observed in other diseases, such as chronic or allergic rhinitis. Application of infliximab (TNFα inhibitor) in an allergic rhinitis mouse model reduced cytokine production and immune cell infiltration into the nasal mucosa[41]. Pirfenidone treatment also successfully treated post-COVID pulmonary fibrosis[42]. Pirfenidone has anti-inflammatory and fibrotic

properties and is used to treat idiopathic pulmonary fibrosis. While its mechanism of action is not entirely understood, it is believed to suppress TNFα and TGFβ production. Moreover, it negatively regulated SMAD and Jak-STAT3 pathways downstream of TGFβ and parallel to PI3K/mTOR signaling[43]. Metformin has been identified as a repurposed drug for atrial fibrillation (AF), a disease characterized by high serum TGFβ and TNFα. Metformin reduces TGFβ production significantly in cardiomyocytes[44]. A recent review highlights metformin's potential to reduce inflammation and improve outcomes in infectious diseases, such as influenza or hepatitis-C[45]. Metformin therapy in diabetes reportedly reduced mortality by roughly 70% after SARS-CoV-2 infection[46], and has indicated successful reduction in the hazard risk of PCS onset[47].

We observed enrichment for the G2M and E2F pathways in the PBS control compared to the TNFα and TGFβ stimulated conditions in the in vitro model. These pathways are closely linked with the initiation and regulation of basal cell differentiation. Assembly of cilium occurs during the G1 phase of the cell cycle, with disassembly taking place during the G2M phase[48], which has a major role in maintaining homeostasis after injury. In addition, the E2F family has been shown to be critical for multiciliogenesis[49,50]. In particular, reduced presence of factor E2F4, shown via a murine model, is associated with fewer ciliated cells in mouse nasal epithelium[51]. Therefore, the reported decreased activity of G2M/E2F pathways indicates a fundamental

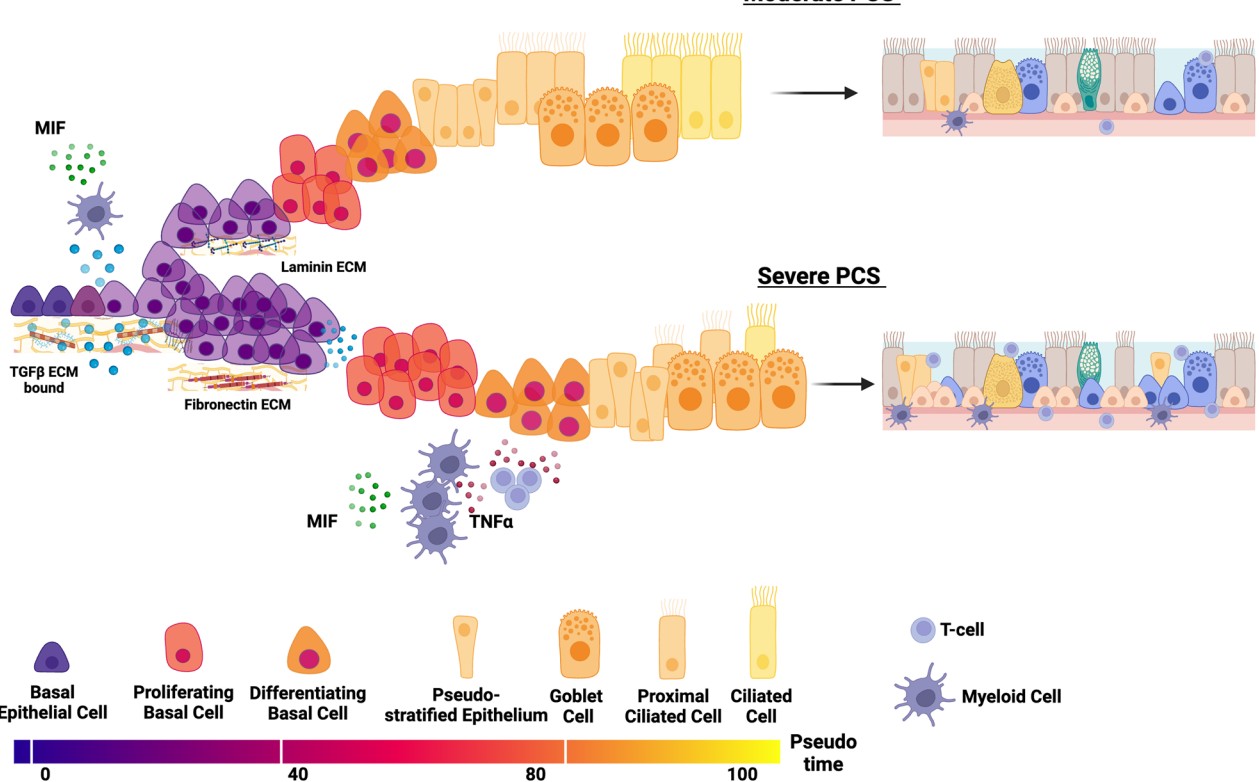

**Fig. 6 | Graphical summary of aberrant molecular and cellular mechanisms in the nasal epithelium in moderate vs severe PCS.** Severe PCS compared to moderate PCS is characterized by an increased abundance of basal epithelial cells due to increased TGFβ signaling. There is also reduced ciliated cell abundance, likely caused by cell-cell interaction between myeloid and T cells with differentiating basal cells with increased TNFα signaling, promoting altered epithelial differentiation. *Created in BioRender.* https://BioRender.com/g6hfbvy.

impedance of normal mucosal repair induced by TNFα and TGFβ, particularly in combination.

A range of clinical interventions have been trialed for both acute SARS-CoV-2 infections and post-COVID syndrome. Peluso & Deeks[52] provide a detailed overview of PCS and therapeutic development. The ALI media used in the in vitro model contains immunomodulatory factors such as hydrocortisone and insulin[53,54]. Dysregulation of inflammation is shown to alter basal cell differentiation[55]. Although reduced, in our model, TNFα-stimulation still reports differentiation of basal cells to ciliated epithelium, indicating a protective effect of the anti-inflammatory regents present in the ALI media. This differentiation is lost with combined exposure to TNFα and TGFβ. These results support why clinical observations that broad-spectrum anti-inflammatory interventions, rather than pathway-specific therapeutics, appear more effective at mitigating PCS progression[56–58]. As such, this model presents as a tool to elucidate the therapeutic potential of current interventions. However, more complex experimental designs are required to appropriately explore this notion.

Our study has limitations due to a relatively small group size, which makes it difficult to extend our findings to other patients with severe PCS. The current findings include patients who reported asymptomatic to very severe-acute SARS-CoV-2 infections. Therefore, our findings may represent a mixture of a resolved severe-acute COVID-19 episode with post-COVID syndrome. Nonetheless, our results provide a foundation for future studies employing our technique to better differentiate these two unique sequelae. The primary scope of this investigation was a focus on cellular and molecular characteristics between degrees of PCS severity. These findings shed light on the core processes promoting worse outcomes in PCS patients. Although our data effectively elucidates and presents the abnormal PCS immune-epithelial cell communication, confirming potential biomarkers in a larger cohort of nasal PCS biopsies would be advantageous. We did not analyze the follow-up data of our patients, which could reveal the predictive value of our findings.

Our results highlight a potential mechanism for virus-free PCS progression. Using scRNA-seq analysis, we detected ciliated cell reduction, heightened immune cell presence, and enrichment of inflammatory pathways in severe PCS in the nasal mucosa. Notably, these cellular and molecular responses occur in the absence of detectable SARS-CoV-2 viral RNA. This contradicts the prevailing hypothesis that PCS is driven by the persistence of a residual viral load[59,60]. However, this conclusion may be influenced by the sampling location with potential viral load persistence in the lower respiratory tract[61]. Nonetheless, our results indicate that the persistence of infection-like symptoms is potentially driven by an aberrant immune-cytokine-epithelial axis and creates a foundation for novel treatment regimens akin to other chronic inflammatory respiratory diseases.

## Methods
Detailed descriptions of the methods are provided in the supplementary material.

### Study population and sample collection
We obtained nasal biopsies from the anterior and medial to the head of the middle turbinate of participants who had a complete set of metadata ($n = 33$), under endoscopic guidance (Fig. 1). According to ethical board approval, as previously published, samples were collected with written and informed consent from patients. The criteria for patient inclusion were (i) polymerase chain reaction confirmed SARS-CoV-2 infection, (ii) a period of 6-months between the infection and the visit and persistence of COVID-19 symptoms for more than three months, (iii) post-acute infection symptom development, (iv) a

worsening of pre-existing comorbidities, and (v) written and informed consent before biopsy collection, aligning with ethical approval. Patients were excluded if they presented with SARS-CoV-2 reinfection at the time of the interview[8]. Supplementary Fig. S1 contains a breakdown of the number of samples selected from the NAPKON cohort.

## Single-cell RNA-seq analysis

Single-cell RNA sequencing (scRNA-seq) was performed in collaboration with the Singleron Company, Cologne, Germany. A detailed description of the pipeline is provided in the supplementary. Briefly, raw gene expression matrices were generated for each sample by a custom pipeline combining *'kallisto'* (v0.46.1) and *'bustools'* (v0.46.1) using GRCh38 as the human reference. The output-filtered gene expression matrices were analyzed in *'R'* (v4.2.1) with the *'DropletUtils'* (v1.8.0) and *'Seurat'* (v4.3) packages. For comparative analysis, we used the 'scProportion Test' from the R library scProportionTest' to quantify differences in cell abundance. Cell interaction analysis utilized the R package 'CellChat' (v1) to discern and visualize intercellular communication patterns, adhering to the developer's guidelines was used according to the developer's vignette [https://github.com/sqjin/CellChat][18]. Differential pathway enrichment analysis employed the PROGENy database to evaluate 14 signaling pathways' activity between different PCS groups, integrating empirical data from perturbation experiments and using a linear statistical analysis model. Single-cell pseudo-time trajectories were constructed using *Monocle* (v2.6.4). Dampened Weighted Least Squares (DWLS) were utilized for gene expression cellular deconvolution.

## TriNetX cohort selection and analysis of PCS-associated comorbidities

To enable a broader analysis of comorbidities associated with PCS, we retrieved a case and control cohort for post-COVID symptoms according to the International Statistical Classification of Diseases and Related Health Problems (ICD-10) codes from the TriNetX Global Collaborative Network[16].

## Air-liquid Interface (ALI) culture

Four transwell permeable supports (#PID0738600, Corning) were coated with 1 mg/ml collagen IV (#5005, Advanced BioMatrix). Healthy donor nasal epithelial cells (NECs), one female and one male, were purchased from PromoCell (#C-12620, PromoCell), defrosted, and expanded on T25 cell culture flasks coated with collagen IV (#5005, Advanced BioMatrix) and collected using an interdental brush with written and informed consent. These individuals were SARS-CoV-2 negative and had no lasting symptoms of acute infection at least six months prior, presented without any respiratory-associated pathologies, and were non-smokers. Collected brushings were stored in bronchial epithelial growth media (BEGM; #CC3171, Lonza) without penicillin/streptomycin in a 15 ml Falcon tube after washing with phosphate-buffered saline. Once confluency was reached, NECs were seeded in the apical chamber of transwell permeable supports (#PID0738600, Corning) coated with 1 mg/ml collagen IV (#5005, Advanced BioMatrix) with supplemented BEGM in the apical and basal chambers. Once a complete cellular monolayer was formed, the apical chamber media was aspirated to initiate epithelial cell differentiation. At this time point, TGFβ (1 ng/ml) and TNFα (10 ng/ml), either alone or in combination, were added to the basal chamber media, with phosphate-buffered saline (PBS) used as a control. Each stimulation condition was maintained in the basal chamber for the entire 28-day period of differentiation, with media changes three times a week. During media change, the apical chamber was washed with Hank's balanced salt solution (HBSS; #14170112, Life Technologies). After 28 days of differentiation, NECs that had been differentiated with or without TNFα were subsequently shipped to Singleron (Cologne) for scRNA-seq analysis. The Transwell apical chamber containing inserts

containing cells was filled with sCelLiVE Tissue Dissociation Solution (#1190062, Singleron), and shipped at 4 °C overnight the day of collection. A schematic of the experimental setup is provided in Supplementary Fig. S13. The overall schematic of TriNetX clinical, in vivo, and in vitro analysis pipelines is located in Supplementary Fig. S17.

## Ethics statement

All samples were collected with written and informed consent. NAPKON's study protocols have been approved by the institutional review boards/ethics committees of all participating study sites. The study was approved by the ethics committee under approval number D537/20, as part of the NAKON-POP cohort, which is conducted within the framework of the German COVID-19 Research Network of University Medicine

## Reporting summary

Further information on research design is available in the Nature Portfolio Reporting Summary linked to this article.

## Data availability

The ALLIANCE data used in this study are sensitive due to individual patient-level data, which will be available upon reasonable request. scRNA-seq data, for the ALI culture generated in this study, have been deposited in the GEO database under accession number: GSE299529. Source data are provided with this paper.

## Code availability

The code underlying the analyses presented in the manuscript has now been made publicly available and is accessible via GitHub at the following link: https://github.com/YamilMaluje/scRNAseq_post-COVID_syndrome.

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

## Acknowledgements

We warmly thank Franziska Beyersdorf and Alina Fischer for their exceptional technical support. Special thanks to all volunteers who participated in this study. A.F. and H.B. acknowledge computational support from the OMICS compute cluster at the University of Lübeck. This work was supported by grants from the COVID-19 Research Initiative Schleswig-Holstein, the Follow-Up of Respiratory Infections in Schleswig-Holstein (FRISH), the German Center of Lung Research (DZL, Funding No. 82DZL001B6), intramural funding of the Christian-Albrechts-University Kiel, the University of Lübeck, and the Leibniz Lung Center, Research Center Borstel. Funding institutions did not participate in the design and conduct of this study. H.B. acknowledges funding by the Deutsche Forschungsgemeinschaft (DFG, German Research Foundation) under Germany`s Excellence Strategy – EXC 22167-390884018. This study was developed by the German Center of Lung Research (DZL) funded by the Federal Ministry of Education and Research (BMBF), the Bavarian State Office for Health and Food Safety (LGL; K1-2497-GLB-20-V4) of the "Gesund.Leben.Bayern." initiative of the Bavarian State Ministry of Health and Care and by grants from Deutsche Forschungsgemeinschaft RTG2668 (Project A5, Project-ID: 435874434) to C.B.S-W. C.A.J reports grants from Federal Ministry of Education and Research (BMBF) for the German Center for Lung Research (DZL), during the conduct of the study; grants from Federal German Ministry of Education and Research (BMBF ABROGATE), Max Zeller Söhne AG, Else-Kröner-Fresinius-Stiftung, DFG Exzellenzinitiative TUM International Graduate School of Science and Engineering (IGSSE) (JADS-Project: PANORAMA), and from DFG Graduiertenkolleg RTG2668 (Project A1, Project-ID: 435874434) outside the submitted work.

## Author contributions

Study design: A.F., M.W., H.B., T.B., M.L.; Data acquisition: A.F., K.D.R., T.B., R.S., S.W., B.V., A.S, A.B., S.B.B., J.P., J.H., F.B., S.S., W.L., C.A.J., C.B.S-W., G.H., E.v.M., K.F.R., A.M.D., N.M., B.S., M.V.K. M.W.; Data analysis: A.F., Y.M. K.D.R., F.O., M.W., H.B.; Data interpretation: A.F., K.D.R., M.L., F.O., M.W., H.B., A.B., S.B.B.; Statistical analysis: A.F., K.D.R., Y.M., F.O., H.B.; A.F. prepared the initial draft of the manuscript; and all authors critically reviewed the manuscript and approved the final version.

## Funding

## Competing interests

C.B.S-W received personal fees from Sanofi, Leti and grants from Zeller and Allergopharma, outside the submitted work. M.W. reports grants from the COVID-19 Research Initiative Schleswig-Holstein, the Follow-Up of Respiratory Infections in Schleswig-Holstein (FRISH), the German Center of Lung Research (DZL, Funding No. 82DZL001B6), RESPIRE-EXCEL (EU-101169403), intramural funding of the Christian-Albrechts-University Kiel, the University of Lübeck, and the Leibniz Lung Center, Research Center Borstel. Funding institutions did not participate in the design and conduct of this study. The remaining authors declare no conflict of interests.

## Additional information

[1]German Center for Lung Research (DZL), Airway Research Center North (ARCN), Borstel, Germany. [2]Department of Paediatric Pneumonology & Allergology, University Clinical Schleswig-Holstein (UKSH), Lübeck, Germany. [3]Division of Epigenetics of Chronic Lung Diseases, Priority Area Chronic Lung Diseases, Research Center Borstel – Leibniz Lung Center, Borstel, Germany. [4]Division of Medical Systems Biology, Institute of Experimental Dermatology, University of Luebeck, Luebeck, Germany. [5]Department of Otorhinolaryngology, Head and Neck Surgery, Christian-Albrechts-University Kiel and UKSH, Campus Kiel, Germany. [6]Department of Pneumology, Lungen Clinic Grosshansdorf, Grosshansdorf, Germany. [7]Airway Research Center North (ARCN), German Center for Lung Research (DZL), Grosshansdorf, Germany. [8]Leibniz Lung Clinic, Department of Internal Medicine I, University Hospital Schleswig-Holstein Campus Kiel, Kiel, Germany. [9]Institute of Epidemiology, Kiel University, Kiel, Germany. [10]Centre of Allergy and Environment (ZAUM), Technische Universität and Helmholtz Centre Munich, Munich, Germany. [11]Comprehensive Pneumology Centre Munich (CPC-M), Member of The German Centre for Lung Research

(DZL), LMU Munich, Germany. [12]Department of Paediatric Pneumology, Allergology and Neonatology, Hannover Medical School, Hannover, Germany. [13]Biomedical Research in End stage and Obstructive Lung Disease Hannover (BREATH), Member of the German Centre of Lung Research (DZL), Hannover, Germany. [14]Cluster of Excellence RESIST (EXC 2155), German Research Foundation (DFG), Hannover Medical School, Hannover, Germany. [15]Institute for Asthma and Allergy Prevention, Helmholtz Centre Munich, German Research Centre for Environmental Health, Neuherberg, Germany. [16]Department of Pulmonary and Allergy, Dr von Hauner University Children's Hospital, Ludwig Maximilian's University, LMU, Munich, Germany. [17]German Center for Child and Adolescent Health (DZKJ), Dr von Hauner Children's Hospital, LMU, Munich, Germany. [18]Department of Paediatric Respiratory Medicine, Inselspital, University Children's Hospital of Bern, University of Bern, Bern, Switzerland. [19]USKH Diagnostics Center, Universtiy Hospital Schleswig-Holstein, Lübeck, Germany. [20]These authors contributed equally: K. D. Reddy, Y. Maluje. [21]These authors jointly supervised this work: H. Busch, M. Weckmann, A. Fähnrich. ✉e-mail: anke.faehnrich@uksh.de

