## [Transparent Peer Review file · Nature Communications]

scRNA-seq reveals persistent aberrant differentiation of nasal epithelium driven by TNF α and TGF β in post-COVID syndrome

Corresponding Author: Dr Anke Fährlich

Version 0:

Reviewer comments:

Reviewer #1

(Remarks to the Author)

In this manuscript Fährlich et al performed single cell RNA sequencing on nasal tissue from patients with post COVID syndrome to understand potential mechanisms behind the condition. The authors then used in vitro ALI cultures to validate several findings. The data provided is of value as the underlying causes of PCS remain elusive and the topic of 'long COVID' is still very much of interest to the global population.

The authors report an association of severity of PCS to the ciliated epithelium in the dataset and conclude those with severe PCS have deficient cilia which may impact their function. They hypothesise this is driven by chronic exposure to inflammation evidenced by increased TNF α and TGF β expression. Cultures of nasal epithelial cells with and without TNF α were executed to demonstrate the impact of this exposure, where an observed altered abundance of ciliated cells was noted.

I have several comments and suggestions for the authors to improve this manuscript before publication.

1. For over 50k cells of scRNA-seq data, I would expect there to be more detailed annotations of the cells found. For example, the "myeloid-dendritic cells" population is indeed most likely a mixed cluster of myeloid cells, however sub clustering of this population and a more granular annotation of the cells would be more appropriate. If the authors were not going to do further analysis on this population in the latter parts of the manuscript, then it would be fine to leave as a mixed cluster, however the cell communication analysis would benefit from knowing the exact cells involved. There are many automated cell annotation tools now available that the authors could use to help with this.
2. In relation to the last point, the authors do not appear to have found any B cells in their dataset which I find surprising. If this is a true finding, then it should be commented on.
3. The authors mention that they did not detect any viral sequences in their data however how they did this is not stated in their methods.
4. In figure 2d, it is not clear what is being compared to what. Extra labelling would help. For a more robust method to compare differential abundance the authors could consider Milo.
5. The authors state that the ciliated cells were depleted in PCS, however this is not clear in Fig 2C and it is not significant in Fig 2D.
6. For Fig 3A, it would make more sense if the two severities were compared with one another or a healthy control to understand what is normal or up/down. For each pathway, it would be beneficial to see the expression of genes contributing to each.
7. As previously mentioned, it is difficult to interpret the cell communication analysis with mixed clusters of cells.
8. For Fig 3E and 3F, either a key or unabbreviated pathway names would be helpful for the reader to understand.
9. For the ALI validation data, there is no mention of cell annotation or any attempts to look at if the stimulation had an effect on cell phenotype. As well as the pathway analysis, differential gene expression between individual cell types would help further validate if the same genes are effected in their in vitro model as their original dataset.
10. Many of the supplementary tables were missing.

Reviewer #2

(Remarks to the Author)

A. Fähnrich and colleagues performed a transcriptomic study of nasal biopsies collected in patients with either moderate or severe post-COVID syndrome (PCS). The authors report a decrease in ciliated cells and their precursors in severe PCS, accompanied by a concomitant increase in basal cells, T cells, and myeloid dendritic cells. Through bioinformatic analyses of pathway enrichment and cell-cell communication networks, they propose that persisting inflammation driven through the MIF cytokine and propagated by TNF- α signaling induces TGF- β -dependent pathways that inhibit ciliated cell differentiation and promote epithelial to mesenchymal transition. To support this model, the authors treated nasal epithelial cells from healthy donors with TNF- α while the cells were differentiating in air/liquid interface cultures. They observed perturbed epithelial differentiation upon TNF- α treatment, with a decrease in ciliation pathways. Based on these experiments, the authors conclude that TNF- α is a contributing factor to the altered nasal epithelium differentiation in severe PCS compared to moderate PCS.

The findings are intriguing and the analysis of single cell RNAseq data from nasal biopsies is state of the art. However, the study remains preliminary, as it lacks a proper control group constituted of individuals who recovered from COVID. The authors chose to compare patients with severe and moderate PCS (and not "mild PCS", as stated in the abstract). Most of the patients included had fatigue and memory problems, in the moderate as in the severe PCS group. Patients from the two groups also had equivalent frequencies of olfactory dysfunction and differed only by less frequent symptoms and by their overall PCS score (Supplementary Table S1). As the authors do not state whether the patients had been hospitalized or not, and as the minimal duration of symptoms for inclusion was just 3 months, it remains possible that the authors findings reflect the resolution of a severe acute COVID episode, rather than PCS per se.

Another limitation is that strong conclusions are drawn mostly on the basis of gene expression patterns. For instance, the authors state in the third part of Results that "aberrant epithelial composition is driven by TNF- α and TGF- β signaling" (title page 7). The sole mechanistic experiment provided in Fig. 5 is not entirely convincing. It does show that TNF- α treatment perturbs nasal epithelial differentiation, which was already suspected (see for instance PMID 37424240) but does not demonstrate that this is the mechanism at work in PCS. In addition, the effect of TNF- α was measured on a weakly ciliated epithelium (9% of cells in cluster 5; Table S5) and remained moderate (decrease from 9% to 6% of ciliated cells).

Additional points:

- No statistical analysis is provided to support a clear separation between the two patient groups (Table S1).
- The authors state that the 4 patients with mild PCS were excluded from the analysis because they were all male and constituted too small a group. While these points may be valid, the data from these 4 patients may have helped mitigate the lack of a control group.
- In the comparison with healthy donor RNAseq data from the literature, it is not clear why the authors used as reference a bulk RNAseq study rather than a single cell RNAseq study such as those mentioned in Sikkema et al. (ref.15). Because of this choice, the authors had to rely on deconvolution of bulk data to estimate the frequency of ciliated cells, which is less precise. Through deconvolution, they obtain an estimated frequency of ciliated cells in the (presumably) combined moderate and severe PCS group of 25%, which is not consistent with their own measurements of ciliated cell frequencies in the single cell dataset (>42% in the moderate group and >30% in the severe group, Table S4).
- It is not clear why the authors did not ascribe cell types to the RNA seq clusters obtained from ALI cultures (Fig. 5). The reader is left to guess that ciliated cells are probably in cluster 5, and that cluster 4 is probably comprises of a type of basal cells. The nature of clusters 6 and 7 remain unspecified. TNF- α treatment significantly decreases clusters 4, 6, and 7, while the decrease in cluster 5 remains below significance. Thus, TNF- α has a more drastic effect on cells whose nature remains unclear.
- In Supplementary Fig. S12, the upregulation of the TNF- α and TGF- β pathways supposed to drive epithelial pathology in severe compared to moderate PCS is not obvious, compared to changes in other pathways such as WNT or Estrogen.
- In the supplementary information file, tables are interspersed with figures, making it difficult to locate relevant information. For instance, Table S4 can be found between Fig. S8 and S9. The Supplementary Table S3 is even more difficult to find, as it turns out to be in an additional excel file.

Reviewer #3

(Remarks to the Author)

So far, a majority of studies have investigated the role of serological and systemic immune response in long COVID, mainly immune perturbations including inflammatory mediators, antibodies, and total T cell responses. This study by Fähnrich et al contributes another insight to post-COVID syndrome (PCS) literature that in the upper respiratory tract, it is not the remaining SARS-CoV-2 virus, but the local long-lasting inflammation and immune response are responsible for the aberrant phenotype.

Here, the authors are the first to utilize single-cell RNA sequencing from nasal tissue to reveal the cellular landscape and suggest the molecular mechanism of PCS in moderate and severe patients. They show that severe PCS is associated with

aberrant nasal epithelia stratification and composition. In particular, differentiated cell populations such as ciliated cells are decreased while basal, T-cells and myeloid cells abundance are increased. Using well-established methods such as PROGENy pathway compendium, CellChat and Gene Set Enrichment Analysis (GSEA), they reveal that this phenomenon is driven by TNFa and TGFb signalling in immune cell compartment, affecting basal cell differentiation process. Findings from this study suggest that targeting TNFa-TGFb axis might be a potential treatment to restore nasal epithelium and counter PCS similar to other chronic inflammatory diseases.

The authors then confirm their clinical PSC samples data by introducing TNFa stimulation on in vitro human nasal epithelial organoids cultured in air-liquid interface (ALI). Again, utilizing sc-RNA sequencing, they illustrate that TNFa is a causal factor in the reduction of differentiation pathway for ciliated cells similar to the observed decrease in severe PCS samples.

This study sc-RNA sequencing analysis are derived from well-established methods, very comprehensive and sophisticated. However, to draw a decisive confirmation of the role of TNFa and TGFb with the current in vitro basal NEC ALI culture data, it might not be sufficient.

1) Firstly, the title and focus of the study is on the role of TNFa and TGFb, however, the in vitro NEC ALI experiment was conducted only with TNFa stimulation. While it is almost impossible to investigate them separately from clinical samples, the in vitro setting is an excellent opportunity to study them separately in detail. The authors need to address this on NEC ALI with a similar experiment using TGFb stimulation by itself, and TNFa + TGFb co-stimulation. This is because it is well known that TNFa and TGFb by itself can promote or inhibit cell proliferation or differentiation, depending on context. Together, whether they might act synergistically or antagonistically is essential to be seen.

2) Secondly, it is important to note that scRNA sequencing is purely transcriptomic analysis. Hence, it is important to confirm the transcriptomics with phenotypic observations. This can be easily done by quantitative/qualitative analysis of immunofluorescence staining for ciliated, goblet, and basal cells, with or without TNFa and TGFb stimulation.

3) Lastly (this is optional, only perform the experiment if the authors can), to decisively conclude the role of TNFa and TGFb on altered basal cell differentiation and suggest a way for treatment, a kind of rescue experiment should be conducted on the NEC ALI culture. The easiest way is to use TNFR inhibitor and TGFR inhibitor in cultures stimulated with TNFa and TGFb to show that these inhibitors can help rescue the phenotype. If the authors can perform downstream inhibition of the pathway, such as on RIP1 or SMAD, that's even better.

Finally, there are some minor issues on the manuscript that need to be fixed:

_ Fig 3.C and D, the Y-axis should be on the same scale for easier visual comparison.

_ Fig 4.B, there's no UMAP for TGFb2, but it was mentioned in the figure legend.

_ Fig 5.A, the UMAP clusters need to be annotated in the legend.

_ Fig 5 legend mentions the F), but there's no Fig 5.F.

Version 1:

Reviewer comments:

Reviewer #2

(Remarks to the Author)

In this revised version, A. Fähnrich et al. have performed additional experiments to test the effect of TNF-a, TGF-b, or the TNF-a/TGF-b combination on a reconstructed epithelium grown from healthy donor nasal cells. They find that TNF-a alone significantly increases ciliated cell differentiation (Fig. 5C, cross symbol), which is exactly the reverse from the main finding in the original manuscript. This unfortunately illustrates the lack of robustness in the manuscript's conclusions.

Further, the authors did not satisfactorily address my other main concern, which is the lack of a relevant control group. The authors compared two groups of patients who shared the major clinical signs of Post-Covid Syndrome (PCS) and differed only by their clinical score. Therefore, the differences between the two groups cannot really inform on the mechanisms underlying PCS.

Reviewer #3

(Remarks to the Author)

The authors have addressed my previous peer-review comments well and carried out necessary experiment to improve their shortcomings in in vitro NEC ALI experiment. The data from this follow up experiment show a drastic difference among control, TNFa, TGFb, and TNFa+TGFb stimulation on NEC ALI differentiation. The result is very clear hence there is no need for further confirmation on protein level with immunofluorescence.

It would be great if the authors can elaborate a bit more on the G2/M checkpoint and E2F pathways since they are well known to involve in multiciliogenesis and formation of differentiated ciliated cells.

What really intriguing from this data is that BEGM culture medium for NEC ALI contains supplements that are anti-

inflammatory such as Hydrocortisone and Insulin (other supplements such as Epinephrine may promote or inhibit inflammatory depending on context). This may explain why under TNF α or TGF β stimulation by itself (especially TNF α), basal cells are still able to differentiate into ciliated cells and secretory cells to some extents. The fact that combination of TNF α and TGF β stimulation can effectively overcome the anti-inflammatory effect from BEGM medium and disrupt the process might suggest that broad spectrum anti-inflammatory reagent may fail under severe PSC and there is a need for more potent reagents. The authors should include this in their discussion as this is highly clinically relevant.

Reviewer #4

(Remarks to the Author)

I've been asked to comment on the revisions performed to Reviewer #1's concerns.

The authors have addressed the comments raised by Reviewer 1. However, I do have additional comments:

1. The Wilcoxon test in the presto package does not take into account the correlation between single cells from the same patient, leading to pseudoreplication biases (see <https://www.nature.com/articles/s41467-021-25960-2> and <https://www.nature.com/articles/s41586-025-08988-y>). The authors should use either the pseudobulk or the mixed-effects model method for differential expression analysis, and for estimating the log fold changes between groups for gene set enrichment.
2. Why did the authors use both FindIntegrationAnchors and Harmony to integrate the data? Usually only one integration method is used. Did the authors use the FindIntegrationAnchors-modified or the unmodified gene expression values for differential expression analysis and gene set enrichment?
3. In the Supplementary File, the authors wrote "Further details regarding the PROGENy analysis process are provided in the supplement" without actually following up on it.
4. The code has not been provided.

Version 2:

Reviewer comments:

Reviewer #4

(Remarks to the Author)

On the authors' response to my previous comment 1:

The authors should cite Nguyen et al, Nature communications 2023 (I assume this is the paper the authors had meant in their response) in the Supplementary Methods section to justify the use of MAST in the DE analysis.

The authors should also add the pseudobulk DE analysis in their rebuttal to the supplementary materials to fend off potential pseudoreplication concerns, because I believe readers who are familiar with single-cell bioinformatics will likely raise them. I also want to comment that, according to the script DGE_scRNAseq_Condition.R in the Github repository, the authors didn't add batch as a covariate (the line `latent.vars = c("Batch")` was commented out).

We have addressed all comments of the reviewers and provide a point-by-point response to the issues raised during the review process. Changes in the text have been marked in yellow.

Reviewer 1

C1 - For over 50k cells of scRNA-seq data, I would expect there to be more detailed annotations of the cells found. For example, the “myeloid-dendritic cells” population is indeed most likely a mixed cluster of myeloid cells, however sub clustering of this population and a more granular annotation of the cells would be more appropriate. If the authors were not going to do further analysis on this population in the latter parts of the manuscript, then it would be fine to leave as a mixed cluster, however the cell communication analysis would benefit from knowing the exact cells involved. There are many automated cell annotation tools now available that the authors could use to help with this.

Response: Thank you for your valuable comment. We acknowledge that automatic cell annotation offers benefits in terms of speed and objectivity; however, it may lack precision in more complex biological contexts. A manual expert annotation, considered the gold standard for cell annotation as described by Clarke et al. (2021) (<https://doi.org/10.1038/s41596-021-00534-0>), provides a more detailed and flexible approach, although it is time-consuming and can be influenced by subjective factors. In terms of cell populations and subclustering, we initially chose not to pursue further subclustering because our primary focus was on the broader cell populations for this analysis. Upon examining the "myeloid-dendritic cells" cluster, we found that none of the markers were particularly distinctive or definitive for identifying a single, cohesive population of either dendritic cells or monocytes. Notably, markers for macrophages (*CSF1R*) were expressed within this cluster, alongside those specific to myeloid dendritic cells (*CD1C*, *CD68*, *HLA-DR*, see Reviewer Figure 1 and 2). Conversely, the expression of *CD1C*, *CD68* & *HLA-DR*, in the absence of B cells (indicated by no expression of *BCR/IGHD* and *MS4A1/CD20*), is more convincing in differentiating a distinct population towards myeloid-dendritic cells. As suggested by the reviewer, we performed a subclustering analysis focused specifically on the myeloid dendritic cell cluster. Our aim was to identify markers distinguishing myeloid dendritic cells (e.g., *CD1C*, *CD68*, *HLA-DR*) from macrophages (e.g., *CSF1R*). However, as shown in Reviewer Figure 3, the analysis did not reveal clearly separable subpopulations within this cluster. This suggests that the myeloid-dendritic cell cluster likely represents a heterogeneous mixture of dendritic cells, monocytes, and macrophages. We have clarified this point in the revised manuscript (page number 5, line 154–158).

Reviewer Figure 1: Violin plots displaying the log-transformed counts per million (ln-CPM) for six key gene markers (CD1C, CLEC10A, IGHD, CD20 (MS4A1), ITGAM, CSF1R), indicative of macrophage and myeloid dendritic cell identity, across 12 identified cell clusters. The X-axis represents the cell clusters, and the Y-axis represents ln-CPM, reflecting gene expression levels.

Reviewer Figure 2: Violin plots displaying the distribution of log-transformed counts per million (ln-CPM), representing gene expression levels, for eight key marker genes: CD86, MS4A1, ITGAM, CD68, CD14, CD1C, CLEC10A, CSF1R, across 12 identified cell clusters. The X-axis represents the cell clusters, and the Y-axis represents ln-CPM, reflecting gene expression levels.

CLEC10A, and CSF1R. These markers are used to assess the identity and characteristics of macrophages and myeloid dendritic cells. The X-axis in each plot represents 12 distinct cell clusters identified, while the Y-axis indicates ln-CPM.

Reviewer Figure 3: Marker gene expression in a sub-clustered myeloid dendritic cell population. Uniform Manifold Approximation Projection (UMAP) plots of myeloid cell marker gene expression in a specified myeloid dendritic cell cluster. Color intensity represents gene expression levels. The lack of discrete expression patterns shows the absence of distinct subpopulations based on these markers within this cluster.

C2 – In relation to the last point, the authors do not appear to have found any B cells in their dataset which I find surprising. If this is a true finding, then it should be commented on.

Response: Thank you for highlighting this issue. We agree that the lack of B cells in the single-cell sequencing data is unexpected. However, as shown in Reviewer Figures 1 and 2, there was minimal to no expression of key B-cell markers, such as CD20 (MS4A1) and IGHD. Consistently, these genes were neither significantly expressed in the list of differentially expressed genes nor identified among the canonical markers used for cluster annotation, see Supplementary table 2. We recognize that this finding requires a further comment on that, and we have addressed it in the revised manuscript (page number 5 lines 157–158).

C3 – The authors mention that they did not detect any viral sequences in their data however how they did this is not stated in their methods.

Response: We thank the reviewer and apologise for this oversight. We have included the methodology for measuring viral sequence detection in the methods now in the Supplementary file page number 2 line 65-68.

C4 – In figure 2d, it is not clear what is being compared to what. Extra labelling would help. For a more robust method to compare differential abundance the authors could consider Milo.

Response: We thank the reviewer for this helpful comment. To improve clarity, we have added the requested labeling to Figure 2D. This panel shows the log₂ fold change in cell type abundance, comparing severe post-COVID syndrome (PCS) patients to those with moderate PCS. Specifically, a value of 1.5 log₂FC indicates a relative increase in T cells in the severe PCS group. We have attempted to clarify this by stating the comparison more directly on the figure and in the text (page number 6 lines 170-176). As suggested by the reviewer, we performed a differential abundance analysis using Milo to further validate our findings. The results were consistent with those obtained via DA-seq (see Reviewer Figure 4). Reviewer Figure 4A displays the UMAP embedding with an overlaid Milo graph, highlighting differentially abundant neighborhoods colored by log-fold change—positive values indicating enrichment in moderate PCS and negative values in severe PCS. Reviewer Figure 4B shows the distribution of these changes across cell clusters. While informative, we believe the redundancy with existing analyses like DA-Seq and the scProportion Test makes these results more appropriate for the reviewer response than for inclusion in the main manuscript or supplement.

Reviewer Figure 4: Differential abundance analysis of cell populations in moderate versus severe conditions using Milo. A) Uniform Manifold Approximation Projection (UMAP) visualization of single-cell RNA sequencing data, colored by cell type (left) and overlaid with a Milo graph illustrating neighborhood overlap size and log fold change (right). Node size in the Milo graph corresponds to the neighborhood size, edge thickness represents the overlap size between neighborhoods, and node color indicates the log fold change in abundance between the moderate PCS and severe PCS (red indicates higher abundance in severe, blue indicates higher abundance in moderate). B) Dot plot showing the distribution of differential abundance log fold change in each cell cluster, as calculated by Milo.

C5 – The authors state that the ciliated cells were depleted in PCS, however this is not clear in Fig 2C and it is not significant in Fig 2D.

Response: We agree with the reviewer that in the original figure this conclusion was unclear. We have made changes to appropriately reflect the relative change in ciliated, proximal ciliated cells rather than bulk ciliated cells in Figure 2D. In addition, we have highlighted this change in the page number 6 at lines 169 - 176.

C6 – For Fig 3A, it would make more sense if the two severities were compared with one

another or a healthy control to understand what is normal or up/down. For each pathway, it would be beneficial to see the expression of genes contributing to each.

Response: We acknowledge the reviewer's note that a comparison to a healthy control would be highly valuable. However, this type of control sample is very rare and was not able to be collected in the same manner as these samples. We have included a healthy control dataset (Supplementary Figure S9) as a point of reference for the nasal cell composition of healthy samples. Nonetheless, moderate and severe PCS samples were compared in Figure 3A, with blue indicating relatively more cells in moderate patient samples.

C7 – As previously mentioned, it is difficult to interpret the cell communication analysis with mixed clusters of cells.

Response: We appreciate the reviewer's comment and have addressed this concern in our response to Comment 1 (C1). As noted, the cluster likely represents a heterogeneous population of dendritic cells, monocytes, and macrophages. This clarification has been included in the revised manuscript (page number 5, line 154–158).

C8 – For Fig 3E and 3F, either a key or unabbreviated pathway names would be helpful for the reader to understand.

Response: We thank the reviewer for this helpful suggestion and agree that providing unabbreviated pathway names enhances clarity and aids reader comprehension. We have therefore added the full pathway names to the legend for Figures 3E and 3F.

C9 – For the ALI validation data, there is no mention of cell annotation or any attempts to look at if the stimulation had an effect on cell phenotype. As well as the pathway analysis, differential gene expression between individual cell types would help further validate if the same genes are affected in their in vitro model as their original dataset.

Response: We appreciate the reviewer's insightful suggestion regarding the need for a more detailed analysis of the ALI validation data. In response, we have now included cell type annotation and examined whether stimulation influenced cell phenotypes. Additionally, we performed differential gene expression analysis within individual cell types to assess whether the same genes were affected in the in vitro model as in the original dataset. We believe these additions have significantly strengthened the validation and provided a more comprehensive understanding of TNF α /TGF β -mediated cellular responses in PCS .

C10 – Many of the supplementary tables were missing.

Response: We sincerely apologise for the confusion regarding the supplementary tables. We have corrected this error and hope that the tables are now more easily located.

Reviewer #2 (Remarks to the Author):

C1 – The authors chose to compare patients with severe and moderate PCS (and not "mild PCS", as stated in the abstract).

Response: We apologise and thank the reviewer for identifying this discrepancy. We have corrected this typographical in the abstract at line 55.

C2 – Most of the patients included had fatigue and memory problems, in the moderate as in the severe PCS group. Patients from the two groups also had equivalent frequencies of olfactory dysfunction and differed only by less frequent symptoms and by their overall PCS score (Supplementary Table S1). As the authors do not state whether the patients had been hospitalized or not, and as the minimal duration of symptoms for inclusion was just 3 months, it remains possible that the authors findings reflect the resolution of a severe acute COVID episode, rather than PCS per se.

Response: We appreciate the reviewer's detailed inspection of this preliminary analysis. We agree that without the additional hospitalisation information, conclusions remain difficult to ascertain. However, an inclusion criterion for these patients was that samples were, in fact, collected 6 months post the acute infection process, by definition, this places these patients into a post-COVID syndrome diagnosis. This point has been clarified in more detail in the text (page number 11 lines 387-391). Nonetheless, due to the current state of PCS research (lacking biomarkers) the reviewer's assertion that the included patients are resolving a severe acute COVID episode is still possible. To accommodate this perspective we have included a discussion regarding this point in our manuscript (page number 11 lines 358 - 362). This addition certainly provides a greater and more well-rounded interpretation of our findings for the broader research community.

C3 – Another limitation is that strong conclusions are drawn mostly on the basis of gene expression patterns. For instance, the authors state in the third part of Results that "aberrant epithelial composition is driven by TNF-a and TGF-b signaling" (title page 7). The sole mechanistic experiment provided in Fig. 5 is not entirely convincing. It does show that TNF-a treatment perturbs nasal epithelial differentiation, which was already suspected (see for instance PMID 37424240) but does not demonstrate that this is the mechanism at work in PCS. In addition, the effect of TFN-a was measured on a weakly ciliated epithelium (9% of cells in cluster 5; Table S5) and remained moderate (decrease from 9% to 6% of ciliated cells).

Response: We thank the reviewer for their highly detailed comment. We acknowledge that more detailed analysis of our in vitro validation model was required, as was mentioned by the other two reviewers. As such, we have conducted a more detailed/intensive model of both TNF α and TGF β , to better resolve the potential synergistic/antagonistic effects. We have coupled this model with more thorough bioinformatic analyses to more appropriately explore the fundamental role of these prominent cytokines. In total, this analysis has certainly elevated the quality of our investigation and has allowed us to generate more concrete hypotheses regarding pathological processes occurring in severe PCS vs moderate PCS.

Additional points: C5 – No statistical analysis is provided to support a clear separation between the two patient groups (Table S1).

Response: We apologize for the lack of clarity regarding the statistical analysis. Chi-squared tests were performed to compare each variable between the moderate and severe PCS groups. We have now revised Supplementary Table S1 to clearly reflect this, including a dedicated column indicating statistical significance.

C6 – The authors state that the 4 patients with mild PCS were excluded from the analysis because they were all male and constituted too small a group. While these points may be valid, the data from these 4 patients may have helped mitigate the lack of a control group.

Response: We agree with the reviewer and also considered this mild PCS group as a solution to mitigate this issue. While the group was incorporated into the analysis, the findings were inconsistent, potentially due to confounding factors such as sex and a limited sample size. Specifically, certain cell populations, such as epithelial deuterosomal, and Mucous cells, had fewer than 20 cells. This low cell count restricts the robustness of subsequent analyses. We hope that in future analyses we would be able to expand the number of patients included in the mild PCS group and better resolve pathological differences in PCS severity.

C7 – In the comparison with healthy donor RNAseq data from the literature, it is not clear why the authors used as reference a bulk RNAseq study rather than a single cell RNAseq study such as those mentioned in Sikkema et al. (ref.15). Because of this choice, the authors had to rely on deconvolution of bulk data to estimate the frequency of ciliated cells, which is less precise. Through deconvolution, they obtain an estimated frequency of ciliated cells in the (presumably) combined moderate and severe PCS group of 25%, which is not consistent with their own measurements of ciliated cell frequencies in the single cell dataset (>42% in the moderate group and >30% in the severe group, Table S4).

Response: We thank the reviewer for this important observation. We fully agree that single-cell RNA-seq (scRNA-seq) data would in principle offer a more direct comparison. However, we deliberately chose to use the bulk RNA-seq dataset as a reference in combination with deconvolution for several reasons. Notably, the single-cell data referenced by Sikkema et al. (ref. 15) were generated using the Seq-Well platform (PMID: 28192419), and additional reference datasets were processed using the 10x Genomics v3 system, which employs an emulsion-based droplet microfluidics approach. In contrast, our data were generated using the Singleron platform, which utilizes a microwell-based system in which individual cells are co-captured with barcoded magnetic beads. These platforms differ considerably in key technical aspects, including cell capture methodology, transcript detection sensitivity, and potential biases in cell-type representation. As such, direct integration or comparison of datasets across these platforms may introduce substantial batch effects and technical

variability, potentially confounding downstream biological interpretation (PMID: 33472597, bioRxiv 2024.06.18.599579)

By relying on a well-characterized bulk RNA-seq dataset combined with a robust deconvolution approach, we aimed to minimize such cross-platform discrepancies and ensure consistency in comparative analysis. While we acknowledge that deconvolution has limitations in precision, we believe that, in this context, it provides a more controlled and interpretable reference framework than attempting to align heterogeneous single-cell technologies.

C8 - It is not clear why the authors did not ascribe cell types to the RNA seq clusters obtained from ALI cultures (Fig. 5). The reader is left to guess that ciliated cells are probably in cluster 5, and that cluster 4 is probably comprises of a type of basal cells. The nature of clusters 6 and 7 remain unspecified. TNF-a treatment significantly decreases clusters 4, 6, and 7, while the decrease in cluster 5 remains below significance. Thus, TNF-a has a more drastic effect on cells whose nature remains unclear.

Response: We appreciate the reviewer for identifying this oversight and acknowledge that this can be improved to enable the reader a deeper understanding of the manuscript. We have made these changes accordingly.

C9 – In Supplementary Fig. S12, the upregulation of the TNF-a and TGF-b pathways supposed to drive epithelial pathology in severe compared to moderate PCS is not obvious, compared to changes in other pathways such as WNT or Estrogen.

Response: We thank the reviewer for their comment and observation. We note that WNT and estrogen are important biological pathways. We have chosen to follow-up TNF α and TGF β for their previously reported ability to alter fundamental cellular differentiation. We believe that changes in WNT signalling, in particular, is a result of TGF β effects, as supported by our validation model. In addition, as mentioned in the discussion TGF β and TNF α are highly clinically applicable due to the current availability of biologicals that can block or inhibit their actions. Therefore, we believed it more relevant to follow-up on these targets in the current body of work to enable a translational interpretation.

C10 – In the supplementary information file, tables are interspersed with figures, making it difficult to locate relevant information. For instance, Table S4 can be found between Fig. S8 and S9. The Supplementary Table S3 is even more difficult to find, as it turns out to be in an additional excel file.

Response: We thank the reviewer for their observation and comment. We have adjusted the document to make it better and easier to follow for the reader.

Reviewer #3 (Remarks to the Author):

C1 - Firstly, the title and focus of the study is on the role of TNF α and TGF β , however, the in vitro NEC ALI experiment was conducted only with TNF α stimulation. While it is almost impossible to investigate them separately from clinical samples, the in vitro setting is an excellent opportunity to study them separately in detail. The authors need to address this on NEC ALI with a similar experiment using TGF β stimulation by itself, and TNF α + TGF β co-stimulation. This is because it is well known that TNF α and TGF β by itself can promote or inhibit cell proliferation or differentiation, depending on context. Together, whether they might act synergistically or antagonistically is essential to be seen.

Response: We kindly thank the reviewer for their observation and recognise the need for the title to more appropriately reflect the contents of the manuscript. We have taken on the reviewer's suggestion and redesigned our air-liquid interface (ALI) cellular model using commercial nasal epithelial cells. In response, we have extended our air-liquid interface (ALI) cellular model incorporating stimulation with TNF α and TGF β , both individually and in combination.

On the background of the comments from all the reviewers we have conducted a more detailed analysis of this highly valuable dataset to enable better comparison to the patient cohort. In total, this inclusion has greatly elevated the overall quality of the work.

C2 - Secondly, it is important to note that scRNA sequencing is purely transcriptomic analysis. Hence, it is important to confirm the transcriptomics with phenotypic observations. This can be easily done by quantitative/qualitative analysis of immunofluorescence staining for ciliated, goblet, and basal cells, with or without TNF α and TGF β stimulation.

Response: We appreciate the reviewer's insightful comment. In our study, we prioritized scRNA-seq because of its superior ability to capture cellular heterogeneity, particularly regarding dynamic responses to TNF α and TGF β . Using our extended Air-Liquid Interface (ALI) cell model, we incorporated stimulation with TNF α and TGF β —both individually and in combination—while conducting high-resolution transcriptomic profiling and detailed cellular annotation. This allowed us to identify key signaling pathways that mirror patterns observed in patient-derived samples. The reviewer's suggestion has helped us refine our interpretation and develop a more robust hypothesis regarding the mechanisms that drive the progression of PCS from moderate to severe. While we acknowledge that immunofluorescence provides valuable protein-level confirmation, our current work focuses on the transcriptional landscape at single-cell resolution. We plan to address protein-level validation in future studies, potentially integrating spatial transcriptomics or multiplexed imaging techniques to connect transcriptomic and phenotypic data.

C3 – Lastly (this is optional, only perform the experiment if the authors can), to decisively conclude the role of TNF α and TGF β on altered basal cell differentiation and suggest a way for treatment, a kind of rescue experiment should be conducted on the NEC ALI culture. The

easiest way is to use TNFR inhibitor and TGFR inhibitor in cultures stimulated with TNFa and TGFb to show that these inhibitors can help rescue the phenotype. If the authors can perform downstream inhibition of the pathway, such as on RIP1 or SMAD, that's even better.

Response: We appreciate the reviewer's suggestion and fully agree that further exploration of these pathways, including the effects of TNFR and TGFR inhibition, would be valuable. The reviewer's suggestion to expand the previous ALI model has provided significant insight into the differential roles of TNFa and TGFb in the differentiation of airway epithelial cells. Per the reviewer's comment, we have subsequently included in the discussion a hypothesis for the role of these two cytokine pathways at page number 10 lines 338- 355. This dialogue will open the potential for future studies from ourselves or other researchers to explore the recovery of the nasal epithelial phenotype.

C4 - _Fig 3.C and D, the Y-axis should be on the same scale for easier visual comparison.

Response: We thank the reviewer for identifying this oversight. We have made this change.

C5 - _Fig 4.B, there's no UMAP for TGFB2, but it was mentioned in the figure legend.

Response: We apologise for this typographical error, and it has been corrected.

C6 - _Fig 5.A, the UMAP clusters need to be annotated in the legend.

Response: We agree with the reviewer and apologise for this oversight. We have added these details to the new figures.

C7 - _Fig 5 legend mentions the F), but there's no Fig 5.F.

Response: Once again we apologise for this typographical error. This has been corrected.

We have addressed all comments of the reviewers and provided a point-by-point response to the issues raised during the review process. Changes in the text have been marked in yellow.

Reviewer #2 (Remarks to the Author):

In this revised version, A. Fähnrich et al. have performed additional experiments to test the effect of TNF- α , TGF- β , or the TNF- α /TGF- β combination on a reconstructed epithelium grown from healthy donor nasal cells. They find that TNF- α alone significantly increases ciliated cell differentiation (Fig. 5C, cross symbol), which is exactly the reverse from the main finding in the original manuscript. This unfortunately illustrates the lack of robustness in the manuscript's conclusions.

Response: We would like to thank the reviewer for the very helpful criticism and agree that the effect of TNF- α alone results in a significant increase of ciliated cells. Our scRNA-Seq measurements identified TGF- β and TNF- α as being present and associated with increasing PCS severity. Therefore, we redesigned our air-liquid-interface experiments to identify the contributions of each stimuli (i.e. PBS, TNF- α and TGF- β individually and in combination). Our results show that TGF- β (compared to PBS) reduced the number of ciliated cells by 2.5 fold, while TNF- α (compared to PBS) increased it by 0.5 fold. However, when both stimuli were applied together, there was a remarkably reduction in the level of ciliated cells (approx. 7.5-fold). The combined stimuli also resulted in a strong increase of the number of differentiating basal cells, which aligns with the pseudo-time analysis and relative cell abundance of our patient samples (Fig. 4). We have taken the valuable criticism of the reviewer, and our revised manuscript now focuses on the combined effect of TGF- β and TNF- α , significantly improving the robustness of our conclusions.

Further, the authors did not satisfactorily address my other main concern, which is the lack of a relevant control group. The authors compared two groups of patients who shared the major clinical signs of Post-Covid Syndrome (PCS) and differed only by their clinical score. Therefore, the differences between the two groups cannot really inform on the mechanisms underlying PCS.

Response: We agree with the reviewer that it would have been more desirable to include a control group. NAPKON-POP included recruitment of previous COVID-19 patients with no or mild residual symptoms after six months, the time point of our biopsies. Our study sampled these patients as well, but we deemed numbers ($n=4$) and cohort composition (only males) not suitable to serve as a reliable control group. Male participants seem to be less affected by higher post-infectious symptoms (PMID: 35875815) making generalisable assumptions on the origin of post-COVID pathways more difficult. Instead, we designed our study to investigate the molecular and cellular differences across the PCS spectrum, moderate versus severe disease manifestations, based on a well-defined clinical classification. While both groups shared PCS diagnosis, we could clearly discern immunological and epithelial state differences associated with increased clinical severity, important for therapeutic stratification. In addition, we provide a reference on typical levels of ciliated cells in healthy and asthmatic participants from the All-Age-Asthma cohort. Shifts in ciliated cell frequencies have been described in asthma (reviewed by Heijink et al. 2020 PMID: 32460363)) but with little change between asthma severity compared to our observation in PCS (Boomer et al. 2024, PMID: 38935626). In

addition, as supported by our ALI model, the mechanisms we have elucidated via our *in vitro* findings leading to clear and testable clinical hypotheses, even in the absence of a control probands.

Reviewer #3 (Remarks to the Author):

The authors have addressed my previous peer-review comments well and carried out necessary experiment to improve their shortcomings in in vitro NEC ALI experiment. The data from this follow up experiment show a drastic difference among control, TNFa, TGFb, and TNFa+TGFb stimulation on NEC ALI differentiation. The result is very clear hence there is no need for further confirmation on protein level with immunofluorescence.

We thank the reviewer for the supportive and constructive feedback. We are pleased that the additional NEC ALI experiments have addressed the reviewer's concerns and that the observed effects of TNF- α , TGF- β , and their combination on epithelial differentiation are now clearly demonstrated. We appreciate the reviewer's view that the observed phenotypic differences are sufficiently robust. We will address the remaining raised suggestions and questions point by point.

C1. It would be great if the authors can elaborate a bit more on the G2/M checkpoint and E2F pathways since they are well known to involve in multiciliogenesis and formation of differentiated ciliated cells.

Response: We appreciate the reviewer's feedback on this important aspect. We have expanded our description of the G2/M checkpoint and E2F signaling pathways in the revised manuscript (see page 8, lines 276-278, and page 10, lines 358-366).

C2: What really intriguing from this data is that BEGM culture medium for NEC ALI contains supplements that are anti-inflammatory such as Hydrocortisone and Insulin (other supplements such as Epinephrine may promote or inhibit inflammatory depending on context). This may explain why under TNFa or TGFb stimulation by itself (especially TNFa), basal cells are still able to differentiate into ciliated cells and secretory cells to some extents. The fact that combination of TNFa and TGFb stimulation can effectively overcome the anti-inflammatory effect from BEGM medium and disrupt the process might suggest that broad spectrum anti-inflammatory reagent may fail under severe PSC and there is a need for more potent reagents. The authors should include this in their discussion as this is highly clinically relevant.

Response: We agree and appreciate the reviewer's insight into the complex immunodulatory environment present in the model. We have revised the Discussion section to explicitly address this point (page 10, lines 367-377).

Reviewer #4 (Remarks to the Author):

I've been asked to comment on the revisions performed to Reviewer #1's concerns.

The authors have addressed the comments raised by Reviewer 1.

We are pleased that the concerns originally raised by Reviewer #1 are now considered fully addressed, as confirmed by Reviewer #4.

However, I do have additional comments:

C1- The Wilcoxon test in the presto package does not take into account the correlation between single cells from the same patient, leading to pseudoreplication biases (see <https://www.nature.com/articles/s41467-021-25960-2> and <https://www.nature.com/articles/s41586-025-08988-y>). The authors should use either the pseudobulk or the mixed-effects model method for differential expression analysis, and for estimating the log fold changes between groups for gene set enrichment.

We appreciate the reviewer's insightful comment. We want to clarify that we did not use the Wilcoxon test through the presto package for differential expression (DE) analysis for the gene set enrichment analysis (GSEA) presented in Supplementary Figure S12. This was a misstatement in the supplementary methods, which has now been corrected (page 4, lines 150-155).

For the DE analysis, we applied the MAST method as implemented in Seurat's *FindMarkers()* function. MAST employs a hurdle model that accounts for dropout and technical variation in single-cell RNA-seq data. We agree that considering patient-level correlation is essential to avoid pseudoreplication and inflated false discovery rates. Based on benchmarking evidence (Nguyen et al., *Genome Biology* 2023), MAST modeling was among the top-performing methods under moderate sequencing depth and batch effects—conditions matching our dataset. Pseudobulk approaches can perform well in balanced designs but are less reliable with unbalanced samples or strong batch effects. While mixed-effects models are theoretically ideal for addressing patient-level variation, they require a sufficient number of samples per group to avoid instability. Given our sample size, we concluded that MAST was the most robust method for this analysis.

To directly address the reviewer's concern, we additionally performed a pseudobulk DE analysis using Seurat's *AggregateExpression* function and DESeq2, followed by GSEA with a Wilcoxon Mann-Whitney test on log fold changes (see Reviewer Figure S1). This independent approach confirmed the enrichment of TNF α , NF- κ B, and EMT pathways in severe PCS patients (Supplementary Figure S12; page 8, lines 238–242), supporting the robustness of our findings. We have updated the manuscript to clarify our methods. While this additional analysis is informative, we believe its redundancy with makes it more suitable for inclusion in the reviewer response rather than in the main manuscript or supplementary materials.

Reviewer Figure1: A positive statistical mean (x-axis) indicates enrichment in severe PCS, while a negative statistical mean indicates enrichment in moderate PCS. Statistical significance was determined at $p < 0.05$ and respective pathways were shown, $n = 25$.

C2- Why did the authors use both FindIntegrationAnchors and Harmony to integrate the data? Usually only one integration method is used.

Response: Thank you for highlighting this important point. We acknowledge that our use of integration methods was not clearly described in the original supplementary methods. To clarify, we did not apply both FindIntegrationAnchors (Seurat-CCA) and Harmony sequentially, but instead used each method on a separate dataset, based on the dataset's characteristics. For the highly heterogeneous nasal swab biopsy dataset, Seurat's canonical correlation analysis (CCA) via FindIntegrationAnchors was applied to address strong batch effects while preserving biological diversity (Stuart et al., 2019). For the smaller, technically consistent ALI culture dataset, Harmony was chosen for rapid, scalable integration that maintains biological variability (Korsunsky et al., 2019). However, recent benchmarking studies indicate that no single integration method performs optimally across all scenarios (Luecken et al., 2022). Therefore, we selected the integration method best suited to the specific structure of each dataset. We have added this clarification to the revised supplementary methods (page 2, lines 48-56).

C3- Did the authors use the FindIntegrationAnchors-modified or the unmodified gene expression values for differential expression analysis and gene set enrichment?

Response: We confirm that we used the unmodified gene expression values (i.e., the raw or normalized data prior to integration) for both differential expression analysis and gene set enrichment analysis. We are aware that the Seurat integration framework modifies gene expression values on a per-cell basis to align datasets into a shared low-dimensional space. While this is effective for correcting batch effects and enabling clustering or visualization, it does not preserve the original gene expression relationships across cells — which are essential for accurate differential expression testing. Therefore, using the integrated (modified) expression matrix for differential expression or enrichment analyses would almost certainly distort true and biological signals. For these analyses, it is essential to work with unmodified expression data that maintains the original inter-cellular gene expression variability. We hope this clarifies our approach and reasoning. We appreciate the opportunity to elaborate on this important methodological point.

C3- In the Supplementary File, the authors wrote "Further details regarding the PROGENy analysis process are provided in the supplement" without actually following up on it.

Response: We apologize for the oversight concerning the details of the PROGENy analysis in the Supplementary File. We have now rewritten the requested detailed description of the PROGENy analysis process in the Supplementary File to ensure clarity and completeness (page 4, lines 142-146).

C4- The code has not been provided.

Response: We thank the reviewer for pointing this out. The code underlying the analyses presented in the manuscript has now been made publicly available and is accessible via GitHub at the following link: https://github.com/YamilMaluje/scRNAseq_post-COVID_syndrome. We have also included this information in the revised manuscript under the "Code Availability" section.

We have addressed all comments of the reviewers and provided a point-by-point response to the issues raised during the review process. Changes in the text have been marked in yellow.

We thank the reviewer for the supportive and constructive feedback. We are pleased that we have addressed the previous reviewer's concerns.

Reviewer #4 (Remarks to the Author)

On the authors' response to my previous comment 1:

The authors should cite Nguyen et al, Nature communications 2023 (I assume this is the paper the authors had meant in their response) in the Supplementary Methods section to justify the use of MAST in the DE analysis.

Response: Thank you very much for your valuable comment and for pointing out the correct citation. We have now included *Nguyen et al., Nature Communications 2023* in the Supplementary Methods section.

The authors should also add the pseudobulk DE analysis in their rebuttal to the supplementary materials to fend off potential pseudoreplication concerns, because I believe readers who are familiar with single-cell bioinformatics will likely raise them.

Response: Thank you very much, as suggested we included this requested Figure as Figures S18 into the supplementary materials.

I also want to comment that, according to the script `DGE_scRNAseq_Condition.R` in the Github repository, the authors didn't add batch as a covariate (the line `latent.vars = c("Batch")` was commented out).

Response: We appreciate you pointing out the commented-out line `latent.vars = c("Batch")` in the `DGE_scRNAseq_Condition.R` script. You are correct that this line was commented out in the version pushed to the repository. This was an oversight on our part. We did, in fact, test the differential gene expression analysis both with and without the batch covariate correction for the ALI culture experiment and we inadvertently left the line commented out in the final script, we corrected this parameter accordingly. We apologize for any confusion this may have caused.